# HYPER-CONNECTIONS

**Defa Zhu, Hongzhi Huang, Zihao Huang, Yutao Zeng, Yunyao Mao, Banggu Wu,
Qiyang Min, Xun Zhou**
Seed-Foundation-Model Team, ByteDance
`{zhudefa,huanghongzhi.51,huangzihao.notabot,yutao.zeng,`
`maoyunyao.myy,wubanggu,minqiyang,zhouxun}@bytedance.com`

## ABSTRACT

We present hyper-connections, a simple yet effective method that can serve as an alternative to residual connections. This approach specifically addresses common drawbacks observed in residual connection variants, such as the seesaw effect between gradient vanishing and representation collapse. Theoretically, hyper-connections allow the network to adjust the strength of connections between features at different depths and dynamically rearrange layers. We conduct experiments focusing on the pre-training of large language models, including dense and sparse models, where hyper-connections show significant performance improvements over residual connections. Additional experiments conducted on vision tasks also demonstrate similar improvements. We anticipate that this method will be broadly applicable and beneficial across a wide range of AI problems.

## 1 INTRODUCTION

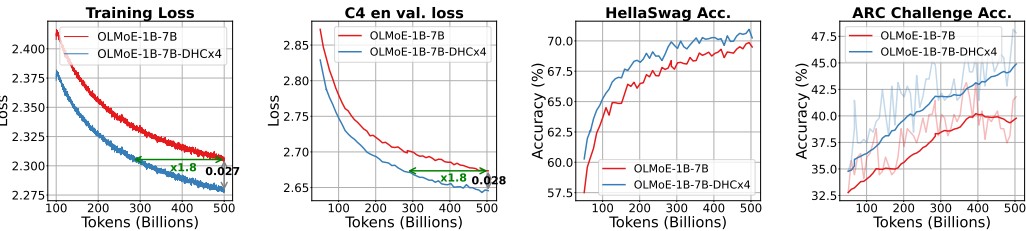

Figure 1: The performance of the baseline model `OLMoE-1B-7B` and the model with hyper-connections, `OLMoE-1B-7B-DHC×4`. **(1)** and **(2)** show the training loss (0.99 EMA smoothed) and the C4-en validation loss, respectively. Our method converges 1.8 times faster compared to the baseline and maintains a significant advantage at the 500B tokens. **(3)** and **(4)** show the accuracy curves on `HellaSwag` and `ARC-Challenge`, demonstrating the superior performance of the `OLMoE-1B-7B-DHC×4` model.

Deep learning has achieved tremendous success across various domains, where residual connections (He et al., 2016) have been instrumental in contemporary neural network architectures, including transformers and CNNs. Residual connections help mitigate the problem of gradient vanishing, enabling the effective training of very deep networks. However, it is important to acknowledge that residual connections are not infallible solutions and still present limitations that remain unresolved.

The two main variants of residual connections, Pre-Norm and Post-Norm, each make distinct trade-offs between gradient vanishing and representation collapse. Pre-Norm applies normalization operations to the input before each residual block, effectively addressing the problem of gradient vanishing (Bengio et al., 1994; Glorot & Bengio, 2010). However, it can also lead to the issue of collapse in deep representations (Liu et al., 2020), where hidden features in deeper layers become highly similar, diminishing the contribution of additional layers as their number increases. In contrast, Post-Norm applies normalization after the output of each residual block, reducing the influence of a hidden state on subsequent layers. This approach can alleviate the issue of representation collapse but

Figure 2: **Hyper-connections (HC) with an expansion rate of** $n = 2$. (a) Residual connections. (b) Hyper-connections: $\beta_1$, $\beta_2$, $\alpha_{0,0}$, $\alpha_{0,1}$, $\alpha_{1,0}$, $\alpha_{1,1}$, $\alpha_{2,1}$, and $\alpha_{2,2}$ are learnable scalars or scalars predicted by the network , depending on the specific HC version. These connections enable lateral information exchange and vertical integration of features across depths. The Transformer with HC is shown in Fig. 17. They can be decoupled into depth-connections and width-connections. (c) Depth-connections perform a weighted sum between the layer output and the hidden vector $h_1$. (d) Width-connections allow information exchange between the hidden vectors $h_1$ and $h_2$.

also reintroduces the problem of vanishing gradients. The vanishing gradient and the representation collapse are like two ends of a seesaw, with these two variants making respective trade-offs between these issues. The key issue is that residual connections, including both Pre-Norm and Post-Norm variants, predefine the strength of connections between the output and input within a layer.

Driven by the limitations of residual connections, an important question arises: *Can neural networks autonomously learn the optimal strength of connections to improve performance?* To address this, we propose hyper-connections (HC), which lead to significantly improved performance with a negligible increase in computation and parameters. We will show that both Post-Norm and Pre-Norm variants can be expressed as specific non-trainable forms of hyper-connections, as discussed in § 3.1.

The core idea of hyper-connections (HC) is to propose learnable *depth-connections* and *width-connections*, as depicted in Fig.2 (b). These connections flexibly integrate features vertically across depths, compared to the residual connections shown in Fig.2 (a). Depth-connections can be considered as a generalized residual connections, assigning weights to the connections between the inputs and outputs of each layer. To enable the network to model different depth-connections simultaneously, we expand the network's input into $n$ copies, each having its own depth connection, as shown in Fig. 2 (b). This design allows multiple hidden vectors to reserve multiple patterns connecting preceding layers, as shown in § 4.5. Moreover, we establish width connections between the $n$ hidden vectors, allowing information exchange between hidden vectors within the same layer, as shown in Fig. 2 (b). We argue that $n\ (> 1)$ hidden states are necessary. As analyzed in Appendix F, the seesaw effect persists when $n = 1$, and experiments show that it does not improve performance, as shown in Fig. 5. In contrast, when $n > 1$, hyper-connections can not only learn to adjust the

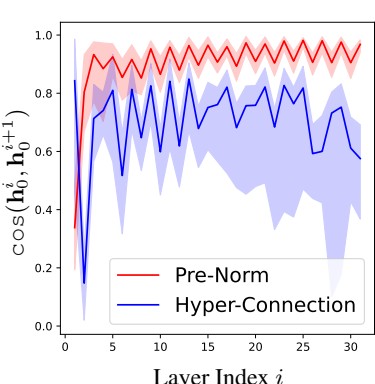

Figure 3: Cosine similarity between the input of the current and the previous layers for the `OLMo-1B` models (Groeneveld et al., 2024). The curve represents the median of similarity, while the shaded area indicates the range between the 5th and 95th percentiles. The red curve shows the model with Pre-Norm, and the blue curve shows that with hyper-connections.

strength of residuals but also rearrange layers, either sequentially or in parallel, as discussed in § 3.2. To further enhance flexibility, we introduce dynamic hyper-connections (DHC), enabling the network to adjust connection weights according to the input. Notably, although HC seem to increase the network's width by $n$ times, the additional parameters and computational cost are almost negligible, as analyzed in Appendix B. The Transformer with HC is shown in Fig. 17.

Our research, primarily centered on large language models (LLMs) pre-training, also extends to visual generation and classification tasks. Using Pre-Norm as a baseline, we demonstrate the significant benefits of hyper-connections, including 1B and 7B dense models as well as 7B MoE models, as

detailed in § 4. The benefits are particularly prominent for OLMoE (Muennighoff et al., 2024) as presented in Fig.1. The model utilizing DHC converges **1.8** times faster and shows an improvement of **6** points on ARC-Challenge compared to the baseline trained with 500 B tokens. According to our visualization analysis, as shown in Fig.3, the baseline model tends toward representation collapse, characterized by high similarity between features of adjacent layers. In contrast, models with HC exhibit significantly lower similarity between features across adjacent layers and a wider range of similarities. This suggests that HC enhance the impact of each layer. Further discussion is provided in §4.5 and in Appendix F. These compelling pieces of evidence demonstrate the generality of the hyper-connections principle, and we anticipate their applicability in numerous other AI challenges.

## 2 METHOD

### 2.1 STATIC HYPER-CONNECTIONS

Consider the hidden vector $\mathbf{h}^{k-1} \in \mathbb{R}^d$ (or $\mathbf{h}^{k-1} \in \mathbb{R}^{d \times 1}$) as the input to the $k$-th layer, with the initial input $\mathbf{h}^0$ to the network. Initially, $\mathbf{h}^0 \in \mathbb{R}^d$ is replicated $n$ times to form the initial *hyper hidden matrix* $\mathbf{H}^0 = \begin{pmatrix} \mathbf{h}^0 & \mathbf{h}^0 & \dots & \mathbf{h}^0 \end{pmatrix}^\mathsf{T} \in \mathbb{R}^{n \times d}$. Here, $n$ is the expansion rate. For the $k$-th layer, the input consists of the hyper hidden matrix from the previous layer $\mathbf{H}^{k-1} = \begin{pmatrix} \mathbf{h}_1^{k-1} & \mathbf{h}_2^{k-1} & \dots & \mathbf{h}_n^{k-1} \end{pmatrix}^\mathsf{T} \in \mathbb{R}^{n \times d}$. Finally, we sum the last hyper hidden matrix row-wise to obtain the required hidden vector, which is then passed through a final projector to produce the final output of the network (i.e., a normalization layer and an unembedding layer in transformers). To simplify the notation in subsequent analysis, we omit the layer index and simply denote the hyper-hidden matrix as $\mathbf{H} = \begin{pmatrix} \mathbf{h}_1 & \mathbf{h}_2 & \dots & \mathbf{h}_n \end{pmatrix}^\mathsf{T}$.

The hyper-connections (HC) can be represented by a matrix $\mathcal{HC}$, where each element defines the connection weight. The matrix is structured as follows:

$$\mathcal{HC} = \begin{pmatrix} \mathbf{0}_{1 \times 1} & \mathbf{B} \\ \mathbf{A_m} & \mathbf{A_r} \end{pmatrix} = \begin{pmatrix} 0 & \beta_1 & \beta_2 & \cdots & \beta_n \\ \alpha_{1,0} & \alpha_{1,1} & \alpha_{1,2} & \cdots & \alpha_{1,n} \\ \alpha_{2,0} & \alpha_{2,1} & \alpha_{2,2} & \cdots & \alpha_{2,n} \\ \vdots & \vdots & \vdots & \ddots & \vdots \\ \alpha_{n,0} & \alpha_{n,1} & \alpha_{n,2} & \cdots & \alpha_{n,n} \end{pmatrix} \in \mathbb{R}^{(n+1) \times (n+1)}. \tag{1}$$

Consider a network layer $\mathcal{T}$, it integrates self-attention layers and feed-forward networks within transformers. The output of the HC, denoted by $\hat{\mathbf{H}}$, can be simply formulated as follows:

$$\hat{\mathbf{H}} = \mathcal{HC}(\mathcal{T}, \mathbf{H}) = \mathbf{B}^\mathsf{T} \mathcal{T}(\mathbf{H}^\mathsf{T} \mathbf{A_m})^\mathsf{T} + \mathbf{A_r}^\mathsf{T} \mathbf{H}. \tag{2}$$

We use $\mathbf{A_m}$ as weights to perform a weighted sum on the input $\mathbf{H} = \begin{pmatrix} \mathbf{h}_1 & \mathbf{h}_2 & \dots & \mathbf{h}_n \end{pmatrix}^\mathsf{T}$ to obtain the input $\mathbf{h}_0$ of the current layer $\mathcal{T}$, which is given by:

$$\mathbf{h}_0^\mathsf{T} = \mathbf{A_m}^\mathsf{T} \mathbf{H}, \tag{3}$$

While $\mathbf{A_r}$ is used to connect $\mathbf{H}$ and map it to a hyper hidden matrix $\mathbf{H}'$, as shown below:

$$\mathbf{H}' = \mathbf{A_r}^\mathsf{T} \mathbf{H}. \tag{4}$$

Subsequently, the output is given by:

$$\hat{\mathbf{H}} = \mathbf{B}^\mathsf{T} (\mathcal{T} \mathbf{h}_0)^\mathsf{T} + \mathbf{H}'. \tag{5}$$

The **depth-connections** can be decoupled as the following matrix, which is shown at Fig 2 (a):

$$\mathcal{DC} = \begin{pmatrix} \mathbf{B} \\ \mathrm{diag}(\mathbf{A_r}) \end{pmatrix} = \begin{pmatrix} \beta_1 & \beta_2 & \cdots & \beta_n \\ \alpha_{1,1} & \alpha_{2,2} & \cdots & \alpha_{n,n} \end{pmatrix} \in \mathbb{R}^{2 \times n}, \tag{6}$$

where the first row $\mathbf{B}$ represents the weights of the output of the current layer $\mathcal{T}$, and the last row $\mathrm{diag}(\mathbf{A_r})$ represents the weights of the input. We use $\mathrm{diag}(\mathbf{A_r})$ to represent the flatten vector of the diagonal entries of $\mathbf{A_r}$.

The **width-connections** matrix can be defined as follows, which is shown at Fig 2 (b):

$$\mathcal{WC} = (\mathbf{A_m} \quad \mathbf{A_r}) \in \mathbb{R}^{n \times (n+1)}. \tag{7}$$

The algorithm that employs hyper-connections is presented in Algorithm 1.

## 2.2 DYNAMIC HYPER-CONNECTIONS

The entries of $\mathcal{HC}$ can dynamically depend on the input $\mathbf{H}$, which the matrix representation of dynamic hyper-connections (DHC) is defined as follows:

$$\mathcal{HC}(\mathbf{H}) = \begin{pmatrix} \mathbf{0}_{1 \times 1} & \mathcal{B}(\mathbf{H}) \\ \mathcal{A}_m(\mathbf{H}) & \mathcal{A}_r(\mathbf{H}) \end{pmatrix} \tag{8}$$

Similarly, given a layer $\mathcal{T}$ and input $\mathbf{H}$, we obtain the output of the DHC as follows:

$$\hat{\mathbf{H}} = \mathcal{HC}(\mathbf{H})(\mathcal{T}, \mathbf{H}). \tag{9}$$

In practice, we combine the dynamic and static matrices to achieve DHC. The dynamic parameters are obtained through a linear transformation. To stabilize the training process, we introduce normalization before the linear transformation and apply the tanh activation function after it, scaling it by a small initial learnable factor. The following equations detail how these dynamic parameters are computed:

$$\overline{\mathbf{H}} = \texttt{norm}(\mathbf{H}) \tag{10}$$

$$\mathcal{B}(\mathbf{H}) = s_\beta \circ \texttt{tanh}(\overline{\mathbf{H}}\mathbf{W}_\beta)^\intercal + \mathbf{B} \in \mathbb{R}^{1 \times n} \tag{11}$$

$$\mathcal{A}_m(\mathbf{H}) = s_\alpha \circ \texttt{tanh}(\overline{\mathbf{H}}\mathbf{W}_m) + \mathbf{A}_m \in \mathbb{R}^{n \times 1} \tag{12}$$

$$\mathcal{A}_r(\mathbf{H}) = s_\alpha \circ \texttt{tanh}(\overline{\mathbf{H}}\mathbf{W}_r) + \mathbf{A}_r \in \mathbb{R}^{n \times n} \tag{13}$$

Our experiments in § 4 demonstrate that dynamic hyper-connections outperform static hyper-connections in language modeling tasks. The PyTorch implementations for both the static and dynamic variants of hyper-connections are detailed in Algorithm 2 and 3.

## 2.3 INITIALIZATION

In order to make the initialization of the hyper-connections equivalent to the Pre-Norm residual connections, we adopt the following initialization strategy. The dynamic parameters $\mathbf{W}_\beta$, $\mathbf{W}_m$, and $\mathbf{W}_r$ in Eqs. 11, 12, and 13 are initialized to 0, while the static matrices are initialized as follows:

$$\begin{pmatrix} \mathbf{0}_{1 \times 1} & \mathbf{B}^k \\ {\mathbf{A_m}}^k & {\mathbf{A_r}}^k \end{pmatrix} = \begin{pmatrix} \mathbf{0}_{1 \times 1} & \mathbf{1}_{1 \times n} \\ \mathbf{e}_{k \bmod n} & \mathbf{e}_{n \times n} \end{pmatrix}, \tag{14}$$

where $k$ is the index of the layer. $\bmod$ denotes the modulo operation.

## 3 WHY HYPER-CONNECTIONS

In this section, we elucidate the rationale behind hyper-connections. We explore how variants of residual connections, namely Pre-Norm and Post-Norm, can be viewed as non-trainable hyper-connections, and introduce the concept of sequential-parallel duality, demonstrating how hyper-connections can dynamically optimize layer arrangements to enhance network performance. A visualize analysis of hyper-connections through an unfolded view is discussed in § 4.5.

### 3.1 RESIDUAL CONNECTIONS AS NON-TRAINABLE HYPER-CONNECTIONS

The Pre-Norm and Post-Norm residual connections can be represented as the following hyper-connections matrices with an expansion rate $n = 1$:

$$\mathcal{HC}_{PreNorm} = \begin{pmatrix} 0 & 1 \\ 1 & 1 \end{pmatrix}, \qquad (15) \qquad \mathcal{HC}_{PostNorm} = \begin{pmatrix} 0 & \frac{1}{\sqrt{\sigma_i^2 + \sigma_o^2 + 2\sigma_{io}}} \\ 1 & \frac{1}{\sqrt{\sigma_i^2 + \sigma_o^2 + 2\sigma_{io}}} \end{pmatrix}, \quad (16)$$

where $\sigma_i$ and $\sigma_o$ denote the standard deviations of the input and output of the neural network layer, respectively, and $\sigma_{io}$ is the covariance between them.

For Pre-Norm, its hyper-connection matrix is a $2 \times 2$ matrix where the bottom right triangular part is filled with $1$ and the rest is a placeholder $0$. For Post-Norm, the weights depend on the variances and covariance of the input and output, forming a $2 \times 2$ matrix. Therefore, their hyper-connection matrices are non-trainable. In this work, we propose hyper-connections that can be $(n+1) \times (n+1)$ matrices, with weights that are trainable or even predicted based on the input. The complete derivation is provided in Appendix G.

### 3.2 Sequential-Parallel Duality

Given a series of neural network modules, we have the option to arrange them either sequentially or in parallel. However, hyper-connections offer an approach that learns to rearrange these layers in a configuration blending both sequential and parallel arrangements.

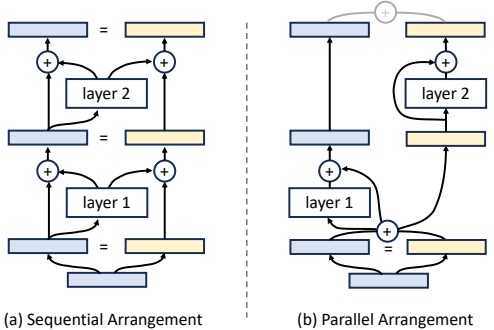

(a) Sequential Arrangement      (b) Parallel Arrangement

Figure 4: Sequential and parallel arrangements of hyper-connections with $n = 2$.

Without loss of generality, we set the expansion rate to $n = 2$. If the hyper-connections are learned as the following matrix, the neural network will be arranged sequentially:

$$\mathcal{HC} = \begin{pmatrix} 0 & 1 & 1 \\ 1 & 1 & 0 \\ 0 & 0 & 1 \end{pmatrix}. \qquad (17)$$

In this case, the depth connection degenerates into a residual connection, as shown in Fig. 4 (a).

When the hyper-connections for odd and even layers (with layer numbering starting from 1) are defined by the following matrices, the neural network will be arranged in parallel every two consecutive layers, similar to the arrangement of parallel transformer blocks in transformers (Wang, 2021), as shown in Fig. 4 (b). The general and complete derivation is provided in Appendix H.

$$\mathcal{HC}_{odd} = \begin{pmatrix} 0 & 1 & 0 \\ 1 & 1 & 1 \\ 1 & 1 & 1 \end{pmatrix}, \qquad (18) \qquad \mathcal{HC}_{even} = \begin{pmatrix} 0 & 0 & 1 \\ 0 & 1 & 0 \\ 1 & 0 & 1 \end{pmatrix}. \qquad (19)$$

Thus, learning the hyper-connection matrix in various forms can create layer arrangements that surpass traditional sequential and parallel configurations, resulting in a soft-mixture or even dynamic

arrangement. For static hyper-connections, the layer arrangement within the network remains fixed after training. In contrast, dynamic hyper-connections allow the arrangement to adapt dynamically for each token.

# 4 RESULTS

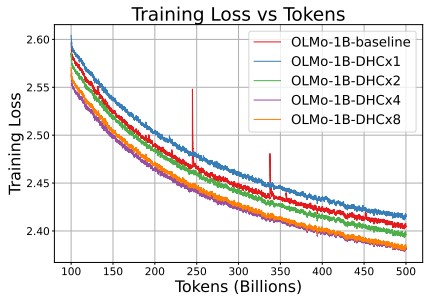 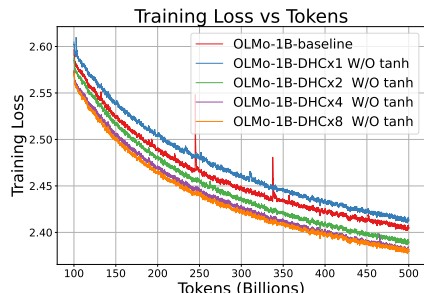

Figure 5: Comparison of training loss curves for different expansion rate. The left subfigure includes models with dynamic hyper-connections (DHC) at various expansion rates, while the right subfigure shows the effect of omitting the tanh function. Both subfigures illustrate how increasing the expansion rate leads to improved training loss performance over 500B tokens. Results are smoothed using an exponential moving average with a coefficient of 0.99.

Table 1: Ablation study on expansion rates $n$ with training on 500 B tokens.

| Methods | V2 Eval Loss ↓ | V2 Eval PPL ↓ | V3 Eval Loss ↓ | V3 Eval PPL ↓ | Down Stream Avg, Acc. ↑ |
|---|---|---|---|---|---|
| OLMo-1B | 2.811 | 18.023 | 2.544 | 14.229 | 62.5 |
| OLMo-1B-DHC×1 W/O tanh | 2.822 | 18.270 | 2.556 | 14.428 | 62.3 |
| OLMo-1B-DHC×2 W/O tanh | 2.792 | 17.663 | 2.537 | 14.033 | 63.8 |
| OLMo-1B-DHC×4 W/O tanh | 2.779 | 17.451 | 2.516 | 13.844 | **64.4** |
| OLMo-1B-DHC×8 W/O tanh | **2.777** | **17.425** | **2.514** | **13.819** | 63.8 |
| OLMo-1B-DHC×1 | 2.819 | 18.125 | 2.556 | 14.418 | 62.3 |
| OLMo-1B-DHC×2 | 2.802 | 17.950 | 2.534 | 14.114 | 63.0 |
| OLMo-1B-DHC×4 | 2.781 | 17.509 | **2.514** | **13.826** | **63.8** |
| OLMo-1B-DHC×8 | **2.778** | **17.445** | 2.516 | 13.843 | 62.8 |

We primarily conduct experiments on pre-training of large language model, including dense and Mixture-of-Experts (MoE) (Shazeer et al., 2017) models, and extend to visual generation and classification tasks. Due to space constraints, we include the vision experiments in the Appendix E.

**Experiment Settings.** We employ the experimental setup outlined by OLMo (Groeneveld et al., 2024) for dense models and by OLMoE (Muennighoff et al., 2024) for MoE models. **For dense models**, we use `dolmap-v1.5-sample` (Soldaini et al., 2024) as our training dataset. We conduct ablation studies on 1B models and assess the effectiveness of our method at the 7B model scale. **For MoE models**, we train the `OLMoE-1B-7B` model, both with and without hyper-connections, on the `OLMOE-MIX` dataset. These models activate 1.3B out of a total of 7B parameters. All experiments are trained on 500B tokens.

**Implementation.** We maintain the training configuration of the baseline model, replacing the residual connections with hyper-connections. The static component in Eqs. 1, 11, 12, 13 does not utilize weight decay, whereas the dynamic component does. Since the hyper hidden vectors of the final transformer block are ultimately summed, we ensure that the standard deviation (`std`) of the output (before the final layernorm and unembedding layers) remains consistent with the original. At initialization, we scale the `std` of the weights of the output module at all layers, including those of the second linear layer of the feedforward network and the output projector of the attention module, by a factor of $\sqrt{n}$,

where $n$ represents the expansion rate. The parameters and computational overhead introduced by hyper-connections is negligible, see Table. 7 and 8.

**Metrics.** In accordance with the methodology of OLMo (Groeneveld et al., 2024), we report the average perplexities (PPL) and losses on both the V2 and V3 validation sets, along with the average metrics for zero-shot evaluation on downstream benchmarks (refer to Table 13). We observe significant volatility in the zero-shot performance indicators for the datasets (highlighted in grey in Table 13), with fluctuations exceeding 20% across neighboring checkpoints. For more reliable and consistent results, we excludes these volatile datasets from our analysis. For the MoE models, in line with OLMoE, we also present losses on V3 validation sets, and accuracies on downstream benchmarks (refer to Table 14).

## 4.1 ABLATION STUDY

We use the dynamic hyperconnections with an expansion rate of $n = 4$ and include the tanh function as the default method, marked with the suffix -DHC, while -SHC denotes static hyper-connections.

The evaluation results are presented in Table 1, and the training loss curves are depicted in Fig. 5. We observe that with an expansion rate of $n = 1$, the performance of DHC is inferior to the baseline. However, for $n > 1$, DHC significantly outperforms the baseline, achieving superior results at $n = 4$, with the increase to $n = 8$ providing minimal additional benefits. Notably, `OLMo-1B-DHC×8 W/O tanh` excels on both V2 and V3 validation sets, with a reduction in `V2 Eval Loss` by **0.034** and `V3 Eval Loss` by **0.029** compared to the baseline. Furthermore, the decline rate of training losses for DHC ($n \geq 2$) is steeper than that of the baseline, and DHC demonstrates greater stability, with no spikes observed in any DHC experiments.

**Static and dynamic hyper-connections.** Table 2 presents an ablation study comparing SHC and DHC. All hyper-connection (HC) variants significantly outperform the baseline. At an expansion rate of 2, the improvements of DHC and SHC are similar. However, at an expansion rate of 4, DHC performs notably better than SHC.

Table 2: Ablation study on static and dynamic hyper-connections with training on 500 B tokens.

| Methods | V2 Eval Loss ↓ | V2 Eval PPL ↓ | V3 Eval Loss ↓ | V3 Eval PPL ↓ | Down Stream Avg, Acc. ↑ |
|---|---|---|---|---|---|
| OLMo-1B | 2.811 | 18.023 | 2.544 | 14.229 | 62.5 |
| OLMo-1B-SHC×2 | 2.799 | 17.778 | 2.538 | 14.152 | 63.4 |
| OLMo-1B-DHC×2 | 2.802 | 17.950 | 2.534 | 14.114 | 63.0 |
| OLMo-1B-DHC×2 W/O tanh | **2.792** | **17.663** | **2.529** | **14.033** | **63.8** |
| OLMo-1B-SHC×4 | 2.791 | 17.671 | 2.528 | 14.025 | 63.6 |
| OLMo-1B-DHC×4 | 2.781 | 17.509 | **2.515** | **13.826** | 63.8 |
| OLMo-1B-DHC×4 W/O tanh | **2.779** | **17.451** | 2.516 | 13.844 | **64.4** |

**The importance of B and $\mathcal{WC}$.** As shown in Table 3, not training $\mathcal{WC}$ leads to significant performance declines, with the V2 loss increasing by **0.021** and the V3 loss by **0.017**, as seen when comparing the 4th and 6th lines of Table 3. In contrast, the impact is less pronounced when B is not trained. Therefore, ensuring the trainability of both $\mathcal{WC}$ and B is crucial.

## 4.2 COMPARISON WITH RELATED WORKS

We implemented the Altup (Baykal et al., 2024) and ResiDual (Xie et al., 2023) methods in OLMo. Altup is motivated to widen the hidden dimension while maintaining low computation cost by passing only a part of hidden state to transformer blocks. By contrast, ResiDual is proposed to combine both Pre- and Post-Norm in a two-stream style. Both methods expand the hidden size by $n$ times with negligible computational overhead, with ResiDual expanding it exactly 2 times. For a fair comparison, we set $n = 2$ in our experiments. Unfortunately, although these methods show gains in the early stages of training, they are gradually surpassed by the baseline, as demonstrated by the results in Table 4 and the training loss curves in Fig. 15.

Table 3: Ablation study on OLMo-1B-DHC×4. In the **B** or $\mathcal{WC}$ column, the symbol "✗" denotes parameters that are not trainable from initialization.

| $\mathcal{WC}$ | **B** | **Tanh** | **V2 Eval Loss ↓** | **V2 Eval PPL ↓** | **V3 Eval Loss ↓** | **V3 Eval PPL ↓** | **Down Stream Avg, Acc. ↑** |
|---|---|---|---|---|---|---|---|
| ✗ | ✓ | ✗ | 2.804 | 17.912 | 2.537 | 14.145 | 62.5 |
| ✓ | ✗ | ✗ | 2.781 | 17.493 | 2.518 | 13.874 | 63.6 |
| ✓ | ✓ | ✗ | **2.779** | **17.773** | **2.516** | **13.823** | **64.4** |
| ✗ | ✓ | ✓ | 2.802 | 17.914 | 2.532 | 14.072 | 63.4 |
| ✓ | ✗ | ✓ | 2.783 | 17.504 | 2.520 | 13.906 | 63.4 |
| ✓ | ✓ | ✓ | **2.781** | **17.835** | **2.515** | **13.807** | **63.8** |

Table 4: Performance of related methods on OLMo-1B models.

| **Methods** | **V2 Eval Loss ↓** | **V2 Eval PPL ↓** | **V3 Eval Loss ↓** | **V3 Eval PPL ↓** | **Down Stream Avg, Acc. ↑** |
|---|---|---|---|---|---|
| OLMo-1B | 2.811 | 18.023 | 2.544 | 14.229 | 62.5 |
| OLMo-1B-ResiDual | 2.825 | 18.375 | 2.551 | 14.346 | 62.0 |
| OLMo-1B-Altup×2 | 2.827 | 18.268 | 2.558 | 14.454 | 62.4 |
| OLMo-1B-DHC×2 | 2.802 | 17.950 | 2.534 | 14.114 | 63.0 |
| OLMo-1B-DHC×2 W/O tanh | **2.792** | **17.663** | **2.529** | **14.033** | **63.8** |

### 4.3 7B MODELS

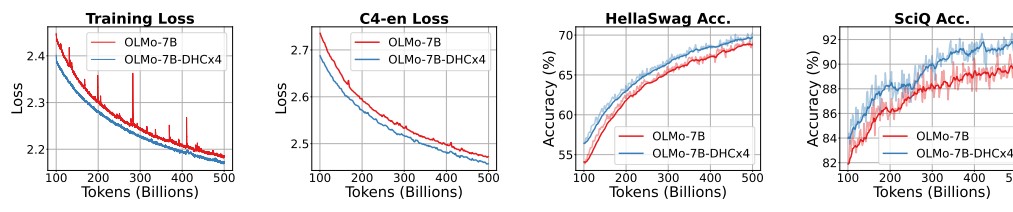

Figure 6: **(1)** and **(2)** Training loss (0.99 EMA smoothed) and C4-en validation loss for `OLMo-7B` and `OLMo-7B-DHC×4` models. **(3)** and **(4)** Accuracy curves on `hellaswag` and `sciq`, demonstrating the superior performance of the `OLMo-7B-DHC×4` model.

We evaluate the effectiveness of hyper-connections on the 7B model, training a model with DHCs with an expansion rate of 4, denoted as `OLMo-7B-DHC×4`. According to Table 5, `OLMo-7B-DHC×4` significantly outperforms the baseline `OLMo-7B` model in all average metrics. In the V2 evaluation, `OLMo-7B-DHC×4` shows improvements of **0.022** for loss and **0.293** for PPL. Furthermore, the average score of downstream benchmarks **0.710** surpasses the baseline 0.701, with the results of specific tasks shown in Fig. 10.

Based on Fig 6, the `OLMo-7B-DHC×4` model consistently shows better metrics compared to baseline, including training and validation loss and accuracy in downstream benchmarks. Notably, after 400 B tokens, the model maintains its improvement without the gains diminishing. This indicates that the `OLMo-7B-DHC×4` model continues to provide consistent benefits in reducing loss, even at higher token counts. Furthermore, according to Fig. 6, the baseline model exhibits frequent spikes, while our model with DHCs shows no spikes throughout the training. This shows that our approach not only achieves better loss but also ensures more stable training.

### 4.4 MoE MODELS

We evaluate the effectiveness of hyper-connections on the Mixture-of-Experts (MoE) model. We retrain the original `OLMoE-1B-7B` model as the baseline and train a model that applies Dynamic

Table 5: Performance of 7B models. FLOPs refers to the computation per token in the forward pass.

| Methods | Params (B) | FLOPs (G) | V2 Loss ↓ | V2 PPL ↓ | V3 Loss ↓ | V3 PPL ↓ | Tasks Avg. Acc. ↑ |
|---|---|---|---|---|---|---|---|
| OLMo-7B | 6.9 | 13.36 | 2.581 | 14.316 | 2.322 | 11.324 | 70.1 |
| OLMo-7B-DHC×4 | 6.9 | 13.38 | **2.559** | **14.023** | **2.304** | **11.120** | **71.0** |

Hyper-Connections (DHC) with $n = 4$, replacing the residual connections. The full results are shown in Fig. 9, which illustrates that hyper-connections outperform residual connections in almost all metrics. In many metrics, our method requires only **half** of the training tokens to achieve the same performance as the baseline. Fig. 1 and Table 6 highlight some of the results, such as a reduction in training loss of approximately **0.027**, a reduction in loss on the C4-en validation set of **0.028**, an improvement of **6** points on the ARC-Challengeand an improvement of **1.2** points on MMLU Var.

Table 6: Downstream evaluations for MoE models training with 500B tokens under the OLMoE evaluation setting. ARC-C stands for ARC-Challenge, and ARC-E for ARC-Easy. MMLU Var is a modified version of MMLU that includes varying few-shot examples, providing stable feedback during early training, as outlined in the OLMoE setting (Muennighoff et al., 2024).

| Methods | MMLU Var | Hella-Swag | ARC-C | ARC-E | PIQA | Wino-Grande | BoolQ |
|---|---|---|---|---|---|---|---|
| OLMoE-1B-7B | 38.5 | 69.5 | 41.8 | 72.8 | 77.6 | 64.4 | 65.4 |
| OLMoE-1B-7B-DHC×4 | **39.7** | **70.2** | **47.8** | **76.7** | **78.2** | **64.6** | **68.5** |

## 4.5 VISUALIZATION ANALYSIS

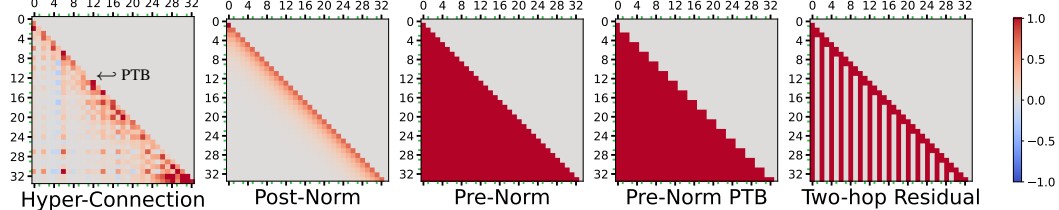

Figure 7: Visualization of connection matrices for hyper-connections and various related baseline methods. The attention layers, which have odd ids, are marked with green tick marks.

In this section, we investigate the learned hyper-connection weights and show how the output of the former layer contributes to the latter ones. To this end, we convert hyper-connections to dense connections cross layers. Consider the input hidden vectors $\mathbf{h}_0^k$ in $k$-th layer, it can be unfolded as a weighted summation over previous layer outputs:

$$\mathbf{h}_0^k = \sum_{j=0}^{k-1} c_{kj}^{(0)} \mathcal{T}^j(\mathbf{h}_0^j),\tag{20}$$

where $c_{kj}^{(0)}$ describes how much layer-$j$ ($\mathcal{T}^j$) contributes to layer-$k$'s input $\mathbf{h}_0^k$. Then, $\mathbf{C}^{(0)}$ denotes a dense connection weight matrix. In particular, let layer-0 be the word embedding and $\mathcal{T}^0$ be an identity mapping, layer-$L+1$ be the hidden state before the unembedding layer, which is a summation over the last hidden vectors, i.e., $\mathbf{h}_0^{L+1} = \sum_j \mathbf{h}_j^L$.

`OLMo-1B-DHC×4` model is adopted for visualization. We take the checkpoint at 500B tokens and forward random validation text to obtain dynamic hyper-connection weights. In addition, we show connection patterns for some related baseline methods. Finally, the visualization is illustrated in Fig. 13. We present the following findings, with more detailed discussions provided in Appendix F.

**Connection patterns for baseline methods.** For Pre-Norm baseline, the connection matrix is simply a lower triangular matrix with diagonal elements erased, because each transformer layer joins the residual equally. In the Pre-Norm parallel transformer block (PTB) baseline, the connection matrix appears jagged because the input to the FFN layer does not depend on the output of the previous attention layer. For Post-Norm baseline, the connection only holds for adjacent layers, as the weight for bottom layers decays every time the residual passes a post-norm layer. For the two-hop residual baseline (Ma et al., 2024), the outputs of attention layers are not added to residual and only contributes to the next one FFN layer, resulting in a vertical strip pattern in the connection matrix.

**$\Lambda$-shaped connection pattern.** In the connection matrix for hyper-connections, a long-term decay pattern can be observed, where layers are generally preferred to rely on a few adjacent layer outputs. Moreover, the bottom layers (e.g. layer 0,2) are observed frequently used in most of subsequent layers. Therefore, the two patterns together form a $\Lambda$-shaped connection pattern. Note that the long-term decay pattern is a Post-Norm style pattern, while the frequently accessed pattern is Pre-Norm style, indicating that the hyper-connection introduces a free mixture of Pre- and Post-Norm architecture.

**Input word embedding is eliminated from model output.** As per the first column in the connection matrix for layer inputs, the input word embedding contributes to most of the layers except for the final one. This last layer, which products the model's output, is used for next token prediction. In most cases, keeping a component of input embedding in model output is harmful to next token prediction, especially when using a tied word embedding such as that employed by `OLMo-1B`. Similar results are found in previous works (Ma et al., 2023).

**Parallel transformer blocks are observed.** As discussed in § 3.2, parallel transformer block, which performs attention and FFN in parallel, is a special case for hyper-connection. In practice, PTB-like patterns, which can be identified by the local jagged pattern, are surprisingly observed to be learned by hyper-connections. For instance, layer 11 has a minimal contribution to the input of layer 12 (refer to row 12 in the hyper-connection connection matrix). This suggests that layers 11 and 12 can operate in parallel, thereby forming a PTB module.

**Attention layers tend to have fewer long-term connections.** It is observed that attention layers at the bottom barely have long-term contribution, a trend that persists until layer 17. Upon examining the connection matrix for hyper hiddens (refer to Fig. 13 in the appendix), it's evident that the outputs of the FFN layers have significantly greater magnitudes than those of the attention layers. This pattern resembles a two-hop residual connection design, wherein the attention output contributes to the input of the following FFN layer, but doesn't join the main residual path.

## 5 RELATED WORK

**Transformers** (Vaswani et al., 2017) have revolutionized various fields, particularly natural language processing and computer vision. They rely heavily on residual connections to facilitate the training of deep models. Our hyper-connections approach can replace residual connections, providing stable training and consistent improvements in both natural language processing and computer vision.

**The issues of gradient vanishing and representation collapse** (Bengio et al., 1994; Glorot & Bengio, 2010; Liu et al., 2020) have been extensively studied. The combinations of normalization techniques (Ioffe & Szegedy, 2015; Ba et al., 2016) and residual connections (He et al., 2016), like Pre-Norm and Post-Norm, actually reflects different emphases in solving these two issues. However, despite these advancements, the fundamental trade-off between gradient vanishing and representation collapse in deep networks remains a critical challenge. Building on these findings, our work introduces a novel approach that enables neural networks to autonomously learn the optimal strength of connections, potentially improving both gradient stability and representation quality.

## 6 CONCLUSION

In conclusion, we have introduced hyper-connections as an effective alternative to residual connections in transformers. Our analysis reveals that hyper-connections not only overcome the limitations of residuals but also enable dynamic adjustments in network architecture. Experimental results confirm their promising benefits across various tasks, including pre-training of large language model, image generation, and image classification.

## ACKNOWLEDGEMENTS

This research was conducted at ByteDance Inc. We are grateful for the suggestions and assistance provided by Yaowei Zheng, Yuyu Zhang, Yunshui Li, Xiang Li, Bairen Yi, Zhenyi Lu and Xintian Han.

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

# A    TRANSFORMER WITH HYPER-CONNECTIONS

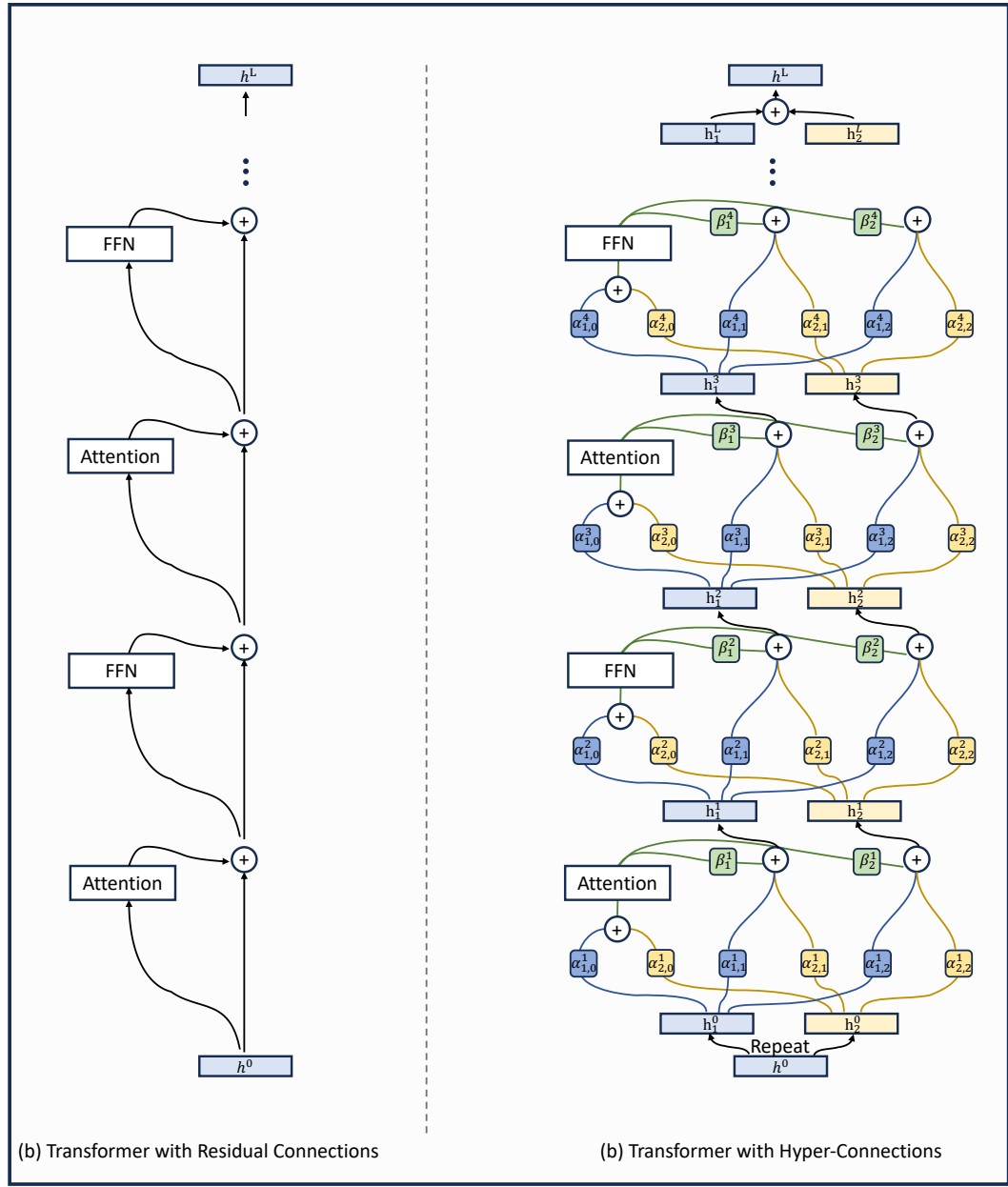

Figure 8: Comparison between transformers with hyper-connections and that with residual connections.

## B    PARAMETERS, COMPUTATION AND MEMORY FOOTPRINT ANALYSIS

**Static Hyper-Connections.** All learnable parameters are included in the hyper-connection matrix $\mathcal{HC}$ in Eq. 1. The number of parameters in one $\mathcal{HC}$ is given by:

$$|\theta_{\text{SHC}}| = |\theta_{\mathbf{B}}| + |\theta_{\mathbf{A}}| = n + n \cdot (n+1) = n \cdot (n+2), \tag{21}$$

where $n$ is the expansion rate, $|\theta_{\mathbf{B}}|$ is the number of parameters in $\mathbf{B}$ in SHC, and $|\theta_{\mathbf{A}}|$ is the number of parameters in $\mathbf{A}$. Each layer contains two hyper-connection modules (one for the self attention and one for the feedforward network). Thus, the number of extra parameters is:

$$P_{\text{extra}} = |\theta_{\text{SHC}}| \times 2 \times L, \tag{22}$$

where $L$ is the number of layers. For example, in OLMo-1B-SHC$\times 4$, $P_{\text{extra}} = 4 \times (4+2) \times 2 \times 16 = 768$.

**Dynamic Hyper-Connections.** The parameters of DHC are defined in Eqs. 10, 11, 12, and 13, and the number of parameters is given by:

$$|\theta_{\text{DHC}}| = |\theta_{\text{norm}}| + |s_{\beta}| + |\theta_{\mathbf{W}_{\beta}}| + |\theta_{\mathbf{B}}| + |s_{\alpha}| + |\theta_{\mathbf{W}_m}| + |\theta_{\mathbf{A}_m}| + |\theta_{\mathbf{W}_r}| + |\theta_{\mathbf{A}_r}| \tag{23}$$

$$= |\theta_{\text{norm}}| + 1 + d_{\text{model}} + n + 1 + d_{\text{model}} + n + d_{\text{model}} \times n + n \times n \tag{24}$$

$$= |\theta_{\text{norm}}| + d_{\text{model}} \times (n+2) + n \times (n+2) + 2, \tag{25}$$

where $d_{\text{model}}$ is the dimension of the hidden states in the transformer, and $|\theta_{\text{norm}}|$ depends on the type of normalization module. In OLMo models, there are no parameters for normalization, so $|\theta_{\text{norm}}| = 0$. In OLMoE, $|\theta_{\text{norm}}| = d_{\text{model}}$. Similar to the static hyper-connections, the number of extra parameters is:

$$P_{\text{extra}} = |\theta_{\text{DHC}}| \times 2 \times L, \tag{26}$$

For example, for OLMo-1B-DHC$\times 4$, $P_{\text{extra}} = (0 + 2048 \times (4+2) + 4 \times (4+2) + 2) \times 2 \times 16 = 394,048$.

The number of parameters for DHC and SHC used in the experiments is detailed in Table 7, while their corresponding FLOPs comparisons are provided in Table 8. Regardless of whether SHC or DHC is used, the additional parameters and computational overhead introduced are minimal and can be considered negligible.

Table 7: Comparison of number of parameters.

| Method | HC Params(B) | Total Params(B) | Total Params $\Delta$ rate (%) |
|---|---|---|---|
| OLMo-1B | - | 1.17676442 | - |
| OLMo-1B-SHC$\times 2$ | 0.0000026 | 1.17676467 | **+0.00002%** |
| OLMo-1B-SHC$\times 4$ | 0.0000077 | 1.17676518 | **+0.00007%** |
| OLMo-1B-DHC$\times 2$ | 0.0002625 | 1.17702688 | **+0.02230%** |
| OLMo-1B-DHC$\times 4$ | 0.0003940 | 1.17715846 | **+0.03349%** |
| OLMo-7B | - | 6.88809574 | - |
| OLMo-7B-DHC$\times 4$ | 0.0013124 | 6.88967027 | **+0.02286%** |
| OLMoE-1B-7B | - | 6.91909427 | - |
| OLMoE-1B-7B-DHC$\times 4$ | 0.0003940 | 6.91948832 | **+0.00570%** |

**Computation Analysis.** The main computational cost of SHC and DHC lies in line 5 of Algorithm 1, where the complexity is $\mathcal{O}(d_{\text{model}} \times n \times (n+1))$. The computational cost of the FFN is $\mathcal{O}(2 \times d_{\text{model}} \times d_{\text{ffn}})$, and that of the projection part of attention is $\mathcal{O}(4 \times d_{\text{model}} \times d_{\text{model}})$. Since $\mathcal{O}(d_{\text{model}} \times n \times (n+1)) \ll \mathcal{O}(4 \times d_{\text{model}} \times d_{\text{model}}) < \mathcal{O}(2 \times d_{\text{model}} \times d_{\text{ffn}})$, the computational cost of HC is negligible compared to the cost of both FFN and the attention projection part. Here, $d_{\text{ffn}}$ is the inner dimension of the FFN. The detailed computation cost statistics are presented in Table 8.

Table 8: FLOPs per token in forward pass.

| Method | HC FLOPs (G) | Total FLOPs (G) | Total FLOPs $\Delta$ rate (%) |
|---|---|---|---|
| OLMo-1B | - | 2.3536 | - |
| OLMo-1B-SHC×2 | 0.0010 | 2.3545 | **+0.038%** |
| OLMo-1B-SHC×4 | 0.0031 | 2.3566 | **+0.127%** |
| OLMo-1B-DHC×2 | 0.0020 | 2.3554 | **+0.076%** |
| OLMo-1B-DHC×4 | 0.0049 | 2.3583 | **+0.200%** |
| OLMo-7B | - | 13.3647 | - |
| OLMo-7B-DHC×4 | 0.0197 | 13.3844 | **+0.147%** |
| OLMoE-1B-7B | - | 2.3580 | - |
| OLMoE-1B-7B-DHC×4 | 0.0049 | 2.3629 | **+0.208%** |

**Memory Footprint.** The introduction of HC results in a minor increase in activation memory usage during training. For a transformer model with $L$ layers, a model dimension of $d_{\text{model}}$, batch size $b$, sequence length $s$, and number of attention heads $a$, the activation memory is calculated as $sbd_{\text{model}}L(34 + 5as/d_{\text{model}})$, as outlined in Korthikanti et al. (2022). Incorporating HC with an expansion rate of $n$ adds an extra memory overhead of $2nsbd_{\text{model}}L$. For $n = 2$, this contributes less than 15% to the total memory usage of a standard transformer. Notably, the memory consumption is mostly driven by the weight parameters, which experience only a slight increase with HC. Additionally, given HC's low computational cost, the hidden states generated by HC can be discarded post forward pass and recomputed during backpropagation to further optimize memory usage. With this approach, the additional memory requirement is reduced to $nsbd_{\text{model}}$. During inference, the memory usage for activations is largely determined by the Key-Value cache, which is not impacted by the extra activations brought by HC. Moreover, the hidden states from earlier layers can be released as soon as the next layer's computations start, significantly lowering memory requirements. The actual memory footprint is empirically measured on 8 GPUs, as shown in Table 9.

Table 9: Measured Memory Footprint on 8 GPUs.

| Method | Memory (GB) | Memory $\Delta$ Rate (%) | Micro Batch Size (tokens per GPU) |
|---|---|---|---|
| OLMo-1B | 41.11 | - | 16,384 |
| OLMo-1B-SHC×2 | 47.55 | **+15.7%** | 16,384 |
| OLMo-1B-SHC×4 | 51.85 | **+26.0%** | 16,384 |
| OLMo-1B-DHC×2 | 47.56 | **+15.7%** | 16,384 |
| OLMo-1B-DHC×4 | 51.86 | **+26.1%** | 16,384 |
| OLMo-7B | 26.27 | - | 2,048 |
| OLMo-7B-DHC×4 | 33.70 | **+28.28%** | 2,048 |
| OLMoE-1B-7B | 31.59 | - | 4,096 |
| OLMoE-1B-7B-DHC×4 | 34.65 | **+9.7%** | 4,096 |

# C   MoE 1B/7B MODEL EXPERIMENTS

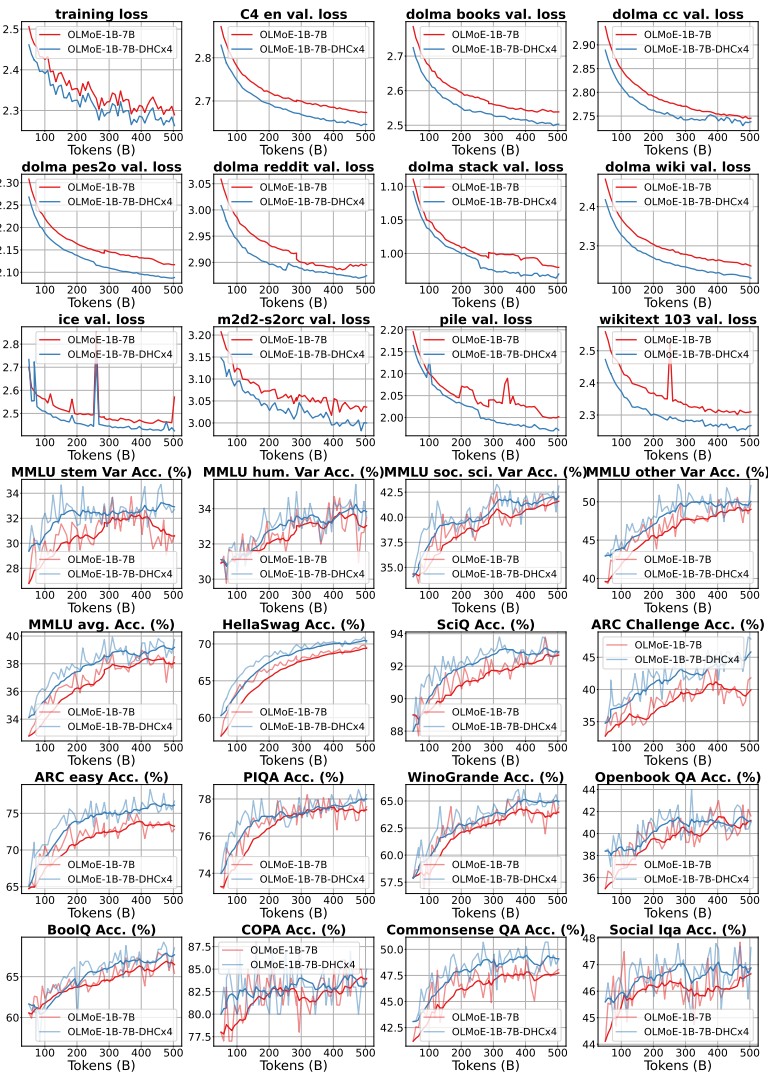

Figure 9: Loss curves in V3 validation sets and accuracy curves on downstream tasks for `OLMoE-1B7B` and `OLMoE-1B7B-DHC×4` models.

# D   7B MODEL EXPERIMENTS

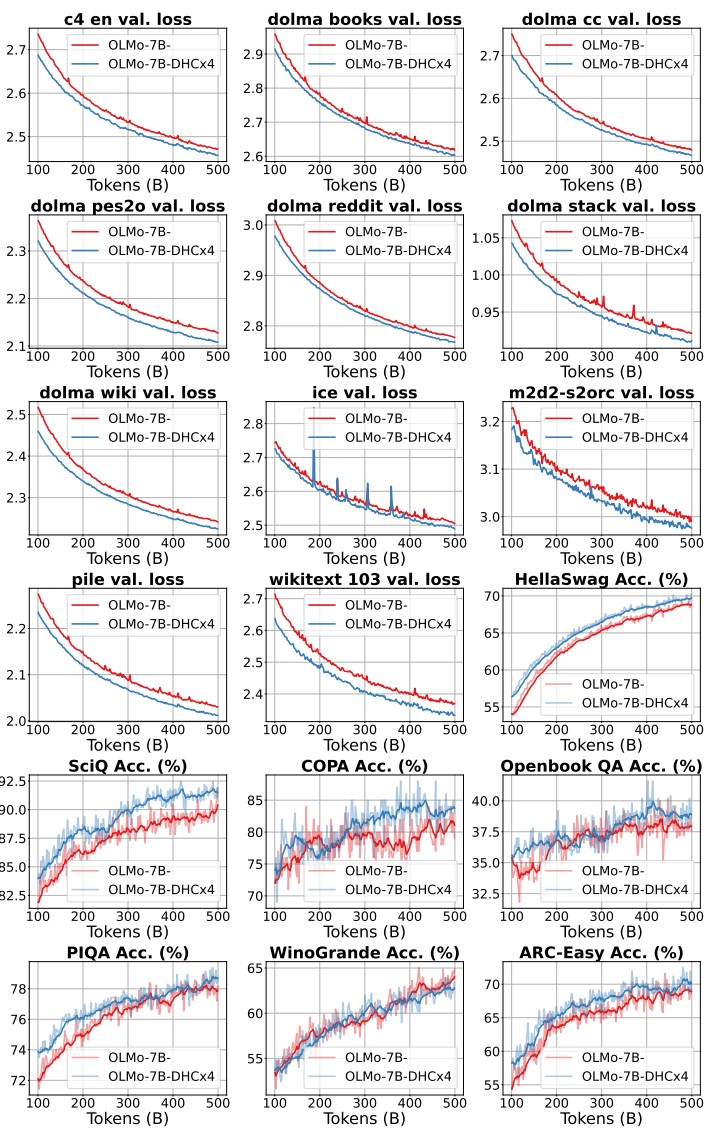

Figure 10: Loss curves in V3 validation set and accuracy curves on downstream tasks for `OLMo-7B` and `OLMo-7B-DHC×4` models.

# E  VISION EXPERIMENTS

**Datasets.** We use the ILSVRC-2012 ImageNet dataset (Deng et al., 2009) with 1k classes and 1.3M images (see ImageNet in the following) for image generation and classification.

## E.1  IMAGE GENERATION

To investigate the generalizability of hyper-connections in image generation, our experiments are conducted using the DiT framework (Peebles & Xie, 2022) training the models for 1400 epochs. In order to save experimental costs, we use FP16 precision, introduce flash-attention to speed up training, and introduce QK-Norm (Wortsman et al., 2023) to stabilize training.

Table 10: Benchmarking class-conditional image generation on ImageNet 256×256, with cfg=1.50. **NP**, **P**, and **R** are short for Numerical Precision, Precision, and Recall, respectively.

| Method | NP | QK-Norm | Size (M) | FID↓ | sFID↓ | IS↑ | P↑ | R↑ |
|---|---|---|---|---|---|---|---|---|
| DiT-XL/2 | FP32 | ✗ | 675 | 2.27 | 4.60 | 278.24 | 0.83 | 0.57 |
| DiT-XL/2 | FP16 | ✓ | 675 | 2.36 | 4.54 | 269.46 | 0.83 | 0.58 |
| DiT-1B/2 | FP16 | ✓ | 983 | 2.13 | 4.50 | 288.69 | 0.82 | 0.59 |
| DiT-XL/2-SHC×2 | FP16 | ✓ | 675 | 2.18 | 4.52 | 287.24 | 0.82 | 0.60 |

Our experimental results demonstrate that DiT models incorporating hyper-connections exhibit comparable performance metrics to DiT models with 50% more parameters. This finding underscores the efficiency and efficacy of hyper-connections in enhancing model performance without increasing model size.

## E.2  IMAGE CLASSIFICATION

For the image classification experiments, we train ViT/16-Base and ViT/16-Large models with images at a resolution of $224 \times 224$ for 300 epochs, following the experimental setup used by (Dosovitskiy et al., 2020).To speed up the training process, we use bfloat16 numerical precision. The training configuration is detailed in Table 12. Within this configuration, we replace the residual connections with static and dynamic hyper-connections, referred to as SHC and DHC, respectively, using an expansion rate of $n = 2$. The top-1 accuracy results are presented in Table 11, and the training loss curves for ViT/16-Large and ViT/16-Large with DHC×2 are shown in Fig. 11.

For the Base model (85M), our re-implemented ViT/16 achieves 76.38% accuracy on $224 \times 224$ images. The SHC and DHC enhance performance to 77.60% and 77.26%, respectively. representing relative increases of **1.22%** and **0.88%**. For the Large model (307M parameters), ViT/16 achieves 77.25% accuracy. The SHC and DHC configurations further enhance accuracy to 78.38% and 79.94%, respectively. This corresponds to relative improvements of **1.13%** and **2.69%**, with DHC showing the highest performance. These results demonstrate that hyper-connections (SHC and DHC) significantly improve accuracy, especially in the Large model scale.

Table 11: Accuracy on ImageNet. **ViT*/16** refers to the results reported by (Dosovitskiy et al., 2020), whereas **ViT/16** denotes our re-implemented baseline. SHC and DHC indicate that residual connections are replaced with static and dynamic hyper-connections, respectively.

| Model Scales | Params (M) | ViT*/16 | ViT/16 | ViT/16-SHC×2 | ViT/16-DHC×2 |
|---|---|---|---|---|---|
| | | $384 \times 384$ | | $224 \times 224$ | |
| **Base** | 85 | 77.91 | 76.38 | **77.60** | 77.26 |
| **Large** | 307 | 76.53 | 77.25 | 78.38 | **79.94** |

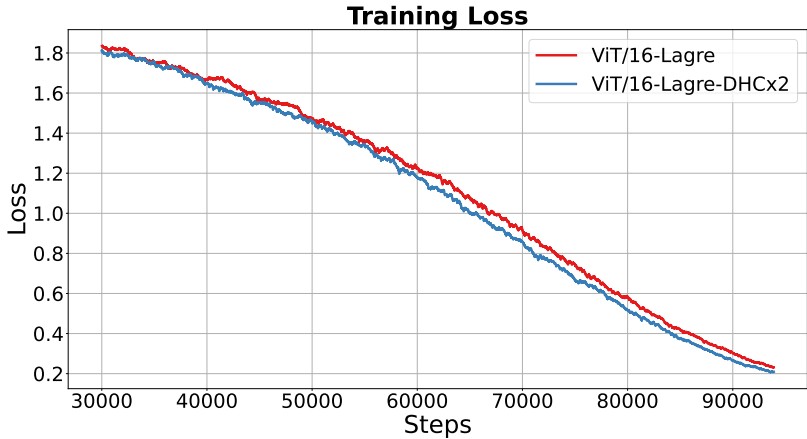

Figure 11: Training loss curves of ViT/16-Large and ViT/16-Large-DHC×2, smoothed using an Exponential Moving Average (EMA) with a decay rate of 0.999. The gain from Hyper-Connections decreases as training progresses, likely due to pass over the same dataset across many epochs, resulting in diminishing returns from the additional capacity provided by Hyper-Connections.

### E.3 VISULIZATION OF DHC

We randomly select three categories from the ImageNet dataset and sample the corresponding examples from the validation set. These samples are fed into the ViT-Base/16-DHC×2 model to compute the dynamic connection weights of the DHC in the final layer. As shown in Fig. 12, we visualize the distribution of these weights. We observe that the intra-class distribution of beta is highly concentrated, indicating that samples within the same category tend to have similar beta values. In contrast, the distribution of alpha is less concentrated, but the differences between the distributions of different categories are more pronounced, as exemplified by $\alpha_{2,0}$.

Table 12: Training hyperparameters for ViT.

| Hyperparameter | Value |
| --- | --- |
| Learning Rate (lr) | 0.003 |
| Batch Size | 4096 |
| Scheduler | Cosine Annealing with Linear Warmup (10k steps) |
| Data Augmentation | Mixup ($\alpha = 0.2$) |
| Epochs | 300 |
| Optimizer | AdamW ($\beta_1 = 0.9$, $\beta_2 = 0.999$, $\epsilon = 1e - 8$) |
| Gradient Clipping | 1.0 |
| Weight Decay | 0.3 |
| Dropout | 0.1 |
| Precision | bf16 |

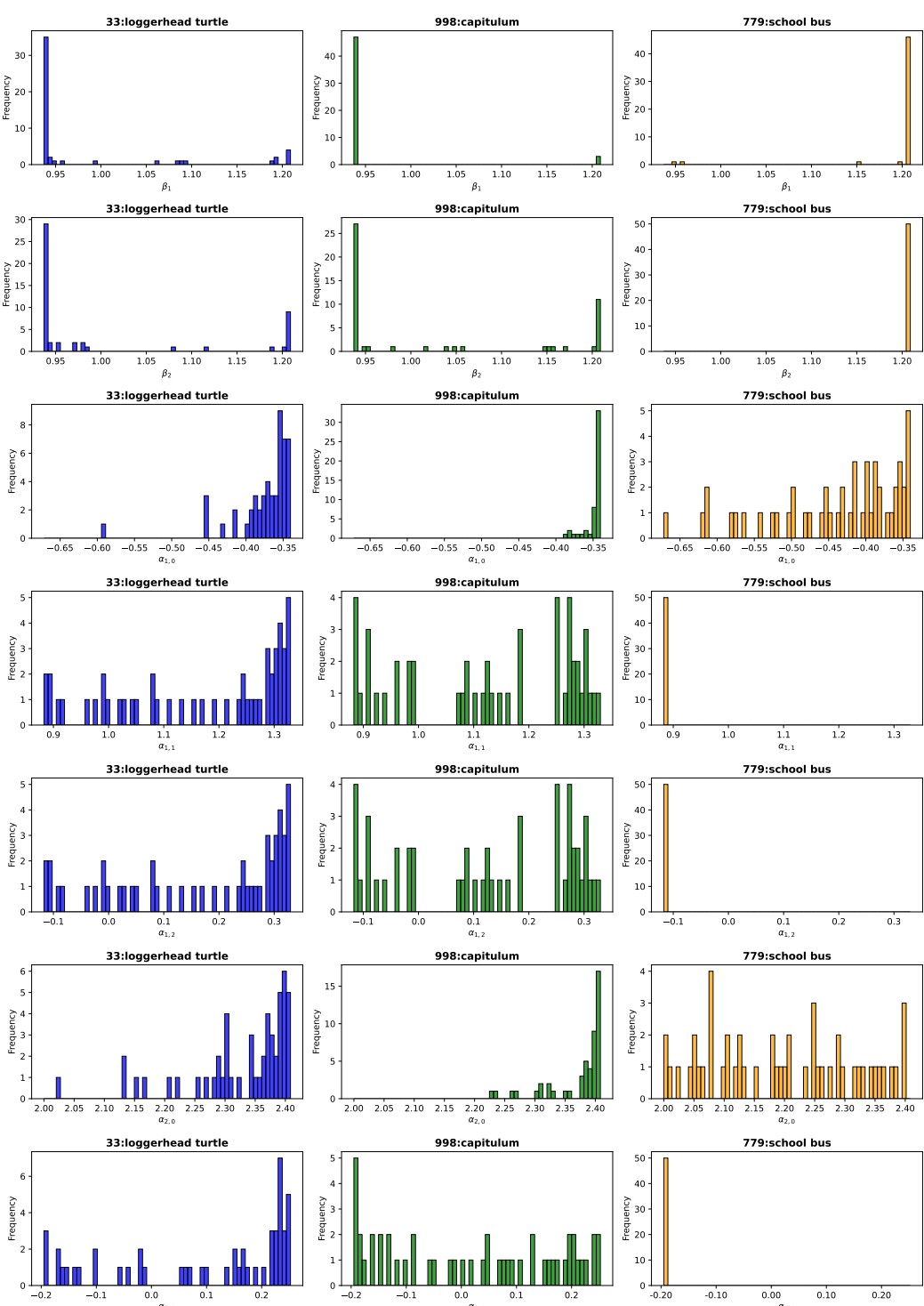

Figure 12: Distribution of weights of last DHC in ViT-Base/16-DHC×2 model.

## F MORE VISUALIZATION AND ANALYSIS

**Unfolding hyper-connections.** We first introduce how to determine the connection matrix $\mathbf{C}^{(0)}$ for hyper-connections. To simplify writing, the layer output $\mathcal{T}^k(\mathbf{h}_0^k)$ is denoted by $\mathcal{T}^k$ for short. The

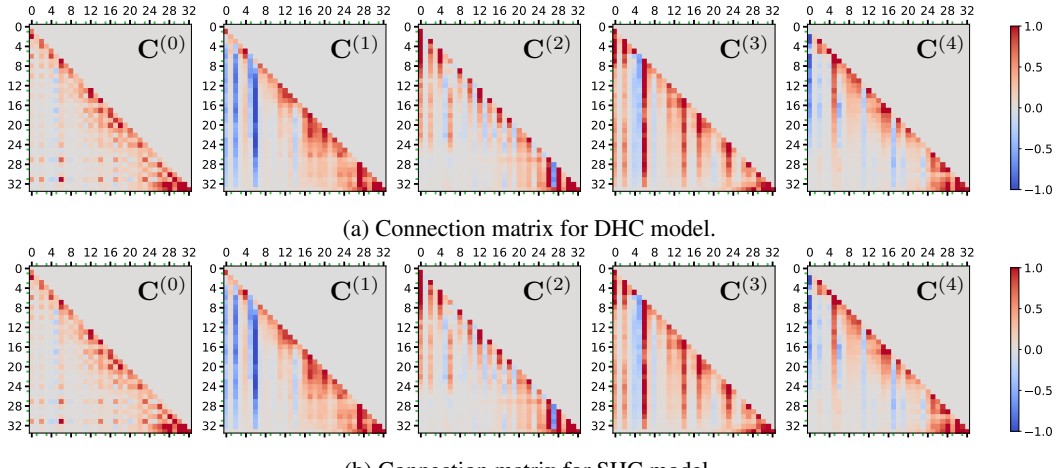

(a) Connection matrix for DHC model.

(b) Connection matrix for SHC model.

Figure 13: **Visualization of unfolded connection matrix.** Matrices from left to right are $\mathbf{C}^{(0)}$(Connections for $\{\mathbf{h}_0^j\}_{j=0}^{L+1}$), $\mathbf{C}^{(i)}$ (Connections for $\{\mathbf{h}'_i{}^j\}_{j=0}^{L+1}$) for $i \in \{1, 2, 3, 4\}$. The attention layers, which have odd ids, are marked with green tick marks.

recurrent form of hyper connection in Eq. 2 is expanded as follows:

$$
\begin{aligned}
\mathbf{h_0}^k &= \mathbf{H}^{k\mathsf{T}}\mathbf{A_m}^k = (\mathcal{T}^{k-1}\mathbf{B}^{k-1} + \mathbf{H}^{k-1\mathsf{T}}\mathbf{A_r}^{k-1})\mathbf{A_m}^k \\
&= \sum_{j=0}^{k-1}\mathcal{T}^j\mathbf{B}^j(\mathbf{A_r}^{j+1}\mathbf{A_r}^{j+2}...\mathbf{A_r}^{k-1})\mathbf{A_m}^k \\
&= \sum_{j=0}^{k-1}\mathcal{T}^j\mathbf{B}^j\left(\prod_{t=j+1}^{k-1}\mathbf{A_r}^t\right)\mathbf{A_m}^k.
\end{aligned}
\tag{27}
$$

Therefore, we obtain connection matrix $c_{kj}^{(0)} = \mathbf{B}^j(\prod_{t=j+1}^{k-1}\mathbf{A_r}^t)\mathbf{A_m}^k$. Similarly, the connection matrix $\mathbf{C}^{(i)}$ for the $i$-th hyper hidden from $k$-th layer can be computed by substituting the last $\mathbf{A_m}^k$ with $\mathbf{A_r}^k$ in Eq. 27, i.e.,

$$
\mathbf{H}'^k = \mathbf{A_r}^{k\mathsf{T}}\mathbf{H}^k = \sum_{j=0}^{k-1}\left(\prod_{t=j+1}^{k}\mathbf{A_r}^t\right)^{\mathsf{T}}\mathbf{B}^{j\mathsf{T}}\mathcal{T}^{j\mathsf{T}}
\tag{28}
$$

$$
c_{kj}^{(i)} = \left(\left(\prod_{t=j+1}^{k}\mathbf{A_r}^t\right)^{\mathsf{T}}\mathbf{B}^{j\mathsf{T}}\right)_i.
\tag{29}
$$

**Visualization for hyper hidden.** We visualize connection matrices for hyper hiddens in Fig. 13 to reveal how hyper-connection maintains intermediate layer outputs. First of all, the four hyper hiddens are dissimilar and show completely different connection patterns. Then, we can see outputs from FFN layers are preserved long-termly in hyper hiddens, while attention layers are reserved less. It is also observed that the long-term connections are usually stored in pairs of hyper hiddens, where the connection is positive in one hyper hidden but negative in the other, for example, column 0 and 2 in $\mathbf{C}^{(1)}$, $\mathbf{C}^{(3)}$. With such strategy, these connections can be easily eliminated in the sum-pooling operation before the unembedding layer.

**SHC shares similar connection pattern with DHC.** We show the connection matrices for `OLMo-1B-SHC×4` model in Fig. 13b. Comparing to DHC, as shown in Fig. 13a, SHC shares exactly the same connection patterns. Moreover, we observe many more PTB-like blocks in SHC, e.g., layers from 13 to 18. Note that the connection relation for SHC is token independent, and such PTB-like blocks can be physically reorganized to be parallelly computed.

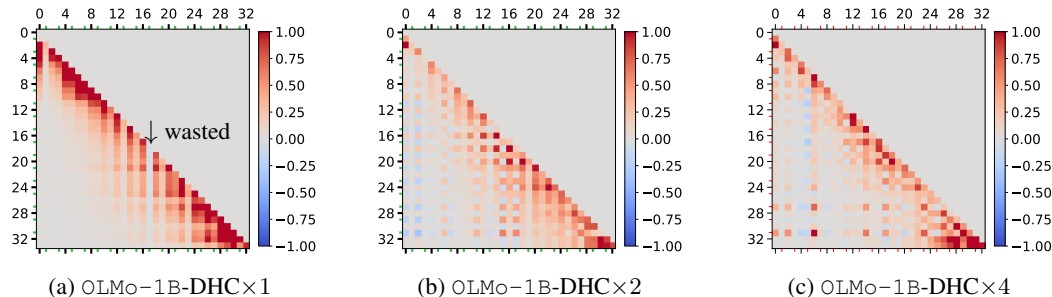

Figure 14: Comparison of unfolded connection matrices for `OLMo-1B`-DHC×1, `OLMo-1B`-DHC×2 and `OLMo-1B`-DHC×4 model.

**How HC×1 fails.** The `OLMo-1B`×1 model is observed to perform worse than baseline in our experiments. Its connection matrix is visualized in Fig. 14 to show how it fails. Above all, we observe that layer 17 is wasted, who has no connection to subsequent layers at all. Secondly, compared to HC×2 and HC×4 models, the Λ shaped pattern does not appear. Note that HC×1 does not support the pattern of Λ in its mathematical formulation, where the connections to previous layers must be weakened or strengthened simultaneously. Thus, the lack of connection from the early layers to the final layers may suffer from gradient vanishing, like post-norm style transformers, which leads to performance degeneration.

## G DERIVATION OF NON-TRAINABLE HYPER-CONNECTION MATRIX FOR RESIDUAL CONNECTIONS

### G.1 PRE-NORM RESIDUAL CONNECTION

In the Pre-Norm residual connection, the input to a layer is first normalized before being passed through the layer. The output of the layer is then added to the original input. This can be represented as:

$$\hat{\mathbf{h}} = \mathcal{T}(\texttt{Norm}(\mathbf{h})) + \mathbf{h}. \tag{30}$$

By incorporating the normalization operator into the layer, $\mathcal{T} := \mathcal{T} \circ \texttt{Norm}$, we can express the entire process as:

$$\hat{\mathbf{h}} = \mathcal{T}(\mathbf{h}) + \mathbf{h}. \tag{31}$$

To express this using hyper-connections, the matrix for Pre-Norm can be structured as follows:

$$\mathcal{HC}_{PreNorm} = \begin{pmatrix} 0 & 1 \\ 1 & 1 \end{pmatrix} \tag{32}$$

Given hyper hidden matrix $\mathbf{H} = \mathbf{h}^{\mathsf{T}}$, we prove that the output of $\mathcal{HC}_{\text{PreNorm}}$ $\hat{\mathbf{H}} = \hat{\mathbf{h}}^{\mathsf{T}}$.

*Proof.*

$$\begin{aligned} \hat{\mathbf{H}} &= \mathcal{HC}(\mathcal{T}, \mathbf{H}) \\ &= \mathbf{B}^{\mathsf{T}} \mathcal{T}(\mathbf{H}^{\mathsf{T}} \mathbf{A_m})^{\mathsf{T}} + \mathbf{A_r}^{\mathsf{T}} \mathbf{H} \\ &= \mathcal{T}(\mathbf{h})^{\mathsf{T}} + \mathbf{h}^{\mathsf{T}} \\ &= \hat{\mathbf{h}}^{\mathsf{T}}. \end{aligned} \tag{33}$$

$\square$

### G.2 POST-NORM RESIDUAL CONNECTION

In the Post-Norm residual connection, the input to a layer is passed through the layer first, and then the output is normalized after being added to the original input. In matrix form, this can be represented as:

$$\mathbf{h}' = \mathcal{T}(\mathbf{h}) \tag{34}$$

The summation of the input and the normalized output of the layer is:

$$\hat{\mathbf{h}} = \texttt{Norm}(\mathbf{h} + \mathbf{h}') \tag{35}$$

We consider Norm to be LayerNorm (Zhang & Sennrich, 2019). The analysis process for RMSNorm is almost identical. In fact, the affine transformation can be incorporated into the subsequent layer, while the mean subtraction operation can be integrated into the current layer.

$$\mathcal{T} = \mathcal{C} \circ \mathcal{T} \circ \mathcal{A}, \tag{36}$$

where $\mathcal{A}$ is the affine transformation, and $\mathcal{C}$ is the re-centering operator. Thus, the mean of the output of $\mathcal{T}$ is 0.

To express this using hyper-connections with an expansion rate $n = 1$, we need a hyper-connection matrix $\mathcal{HC}$ that encapsulates this operation:

$$\mathcal{HC}_{PostNorm} = \begin{pmatrix} 0 & \frac{1}{\sqrt{\sigma_{\mathbf{h}}^2 + \sigma_{\mathbf{h}'}^2 + 2\sigma_{\mathbf{hh}'}}} \\ 1 & \frac{1}{\sqrt{\sigma_{\mathbf{h}}^2 + \sigma_{\mathbf{h}'}^2 + 2\sigma_{\mathbf{hh}'}}} \end{pmatrix} = \begin{pmatrix} 0 & \mathbf{B} \\ \mathbf{A}_m & \mathbf{A}_r \end{pmatrix}. \tag{37}$$

Similar to the previous proof, we prove that the output of $\mathcal{HC}_{\text{PostNorm}}$ is equivalent to the transpose of the output of the Post-Norm residual connection:

$$\hat{\mathbf{H}} = \hat{\mathbf{h}}^{\mathsf{T}}. \tag{38}$$

*Proof.* Note that

$$\sigma_{\mathbf{h}+\mathbf{h}'} = \sqrt{\sigma_{\mathbf{h}}^2 + \sigma_{\mathbf{h}'}^2 + 2\sigma_{\mathbf{h}\mathbf{h}'}}. \tag{39}$$

Given this fact, we can derive the Post-Norm:

$$\begin{aligned}
\hat{\mathbf{h}} &= \text{Norm}(\mathbf{h}' + \mathbf{h}) \\
&= \frac{\mathbf{h}' + \mathbf{h} - \mu_{\mathbf{h}'+\mathbf{h}}}{\sigma_{\mathbf{h}+\mathbf{h}'}} \\
&= \frac{1}{\sigma_{\mathbf{h}'+\mathbf{h}}}(\mathbf{h}' + \mathbf{h}) \\
&= \frac{1}{\sqrt{\sigma_{\mathbf{h}}^2 + \sigma_{\mathbf{h}'}^2 + 2\sigma_{\mathbf{h}\mathbf{h}'}}}(\mathbf{h}' + \mathbf{h})
\end{aligned} \tag{40}$$

For hyper-connections side, we have:

$$\begin{aligned}
\hat{\mathbf{H}} &= \mathbf{B}^{\mathsf{T}}\mathbf{h}'^{\mathsf{T}} + \mathbf{H}' \\
&= \mathbf{B}^{\mathsf{T}}\mathbf{h}'^{\mathsf{T}} + \mathbf{A}_r\mathbf{H} \\
&= \mathbf{B}^{\mathsf{T}}\mathbf{h}'^{\mathsf{T}} + \mathbf{A}_r\mathbf{h}^{\mathsf{T}} \\
&= \frac{1}{\sqrt{\sigma_{\mathbf{h}}^2 + \sigma_{\mathbf{h}'}^2 + 2\sigma_{\mathbf{h}\mathbf{h}'}}}\mathbf{h}'^{\mathsf{T}} + \frac{1}{\sqrt{\sigma_{\mathbf{h}}^2 + \sigma_{\mathbf{h}'}^2 + 2\sigma_{\mathbf{h}\mathbf{h}'}}}\mathbf{h}^{\mathsf{T}} \quad = \hat{\mathbf{h}}^{\mathsf{T}}.
\end{aligned} \tag{41}$$

$\square$

# H SEQUENTIAL-PARALLEL DUALITY

## H.1 HYPER-CONNECTION MATRIX OF SEQUENTIAL ARRANGEMENT

In this section, we demonstrate that the following hyper-connection matrix will produce $n$ identical networks arranged sequentially with residual connections between them:

$$\mathcal{HC} = \begin{pmatrix} \mathbf{0}_{1\times 1} & \mathbf{1}_{1\times n} \\ \mathbf{e}_1 & \mathbf{e}_{n\times n} \end{pmatrix}, \tag{42}$$

where $\mathbf{e}_{n\times n}$ denotes an $n \times n$ identity matrix, $\mathbf{e}_i \in \mathbb{R}^{n\times 1}$ represents the $i$-th column of $\mathbf{e}_{n\times n}$, and $\mathbf{1}_{1\times n}$ signifies a $1 \times n$ matrix of ones.

We will use mathematical induction to prove that $\mathbf{h}_i^k = \mathbf{h}_j^k$ and $\mathbf{h}_i^{k+1} = \mathcal{T}^k(\mathbf{h}_i^k) + \mathbf{h}_i^k$, $\forall i, j \in \{0, 1, \dots, n\}$, $\forall k \in \{0, 1, \dots, L\}$, where $L$ is the number of layers.

*Proof.* BASE CASE

For $k = 0$, we have the initial condition $\mathbf{h}_i^0 = \mathbf{h}_j^0$, $\forall i, j \in \{0, 1, \dots, n\}$, as we define $\mathbf{H}^0 = \begin{pmatrix} \mathbf{h}^0 & \mathbf{h}^0 & \dots & \mathbf{h}^0 \end{pmatrix}^\mathsf{T} \in \mathbb{R}^{n\times d}$.

INDUCTION HYPOTHESIS

Assume that for some $k \in \{1, \dots, L-1\}$, we have $\mathbf{h}_i^k = \mathbf{h}_j^k$ and $\mathbf{h}_i^k = \mathcal{T}^k(\mathbf{h}_i^{k-1}) + \mathbf{h}_i^{k-1}$, $\forall i, j \in \{0, 1, \dots, n\}$.

INDUCTION STEP

We have

$$\mathbf{H}^{k+1} = \mathcal{HC}(\mathcal{T}^k, \mathbf{H}^k) \tag{43}$$

$$= \mathbf{B}^\mathsf{T}(\mathbf{h}_0'^k)^\mathsf{T} + \mathbf{H}'^k \tag{44}$$

$$= \mathbf{B}^\mathsf{T}\mathbf{A_m}^\mathsf{T}\mathbf{H}^k + \mathbf{A_r}^\mathsf{T}\mathbf{H}^k \tag{45}$$

$$= \mathbf{1}_{n\times 1}\mathcal{T}^k(\mathbf{e}_1^\mathsf{T}\mathbf{H}^k) + \mathbf{e}_{n\times n}\mathbf{H}^k \tag{46}$$

$$= \begin{pmatrix} \mathcal{T}^k(\mathbf{h}_1^k) & \mathcal{T}^k(\mathbf{h}_1^k) & \dots & \mathcal{T}^k(\mathbf{h}_1^k) \end{pmatrix}^\mathsf{T} + \begin{pmatrix} \mathbf{h}_1^k & \mathbf{h}_2^k & \dots & \mathbf{h}_n^k \end{pmatrix}^\mathsf{T} \tag{47}$$

$$= \begin{pmatrix} \mathcal{T}^k(\mathbf{h}_1^k) + \mathbf{h}_1^k & \mathcal{T}^k(\mathbf{h}_1^k) + \mathbf{h}_2^k & \dots & \mathcal{T}^k(\mathbf{h}_1^k) + \mathbf{h}_n^k \end{pmatrix}^\mathsf{T} \tag{48}$$

$$= \begin{pmatrix} \mathbf{h}_1^{k+1} & \mathbf{h}_2^{k+1} & \dots & \mathbf{h}_n^{k+1} \end{pmatrix}^\mathsf{T} \tag{49}$$

Since $\mathbf{h}_i^k = \mathbf{h}_j^k$, $\forall i, j \in \{0, 1, \dots, n\}$, it follows that $\mathcal{T}^k(\mathbf{h}_1^k) + \mathbf{h}_i^k = \mathcal{T}^k(\mathbf{h}_1^k) + \mathbf{h}_j^k$. Thus, we have

$$\mathbf{h}_i^{k+1} = \mathbf{h}_j^{k+1} \tag{50}$$

Since $\mathbf{h}_i^k = \mathbf{h}_j^k$, $\forall i, j \in \{0, 1, \dots, n\}$, it follows that $\mathbf{h}_1^k = \mathbf{h}_i^k$, $\forall i \in \{0, 1, \dots, n\}$. Thus, we have

$$\mathbf{h}_i^{k+1} = \mathcal{T}^k(\mathbf{h}_1^k) + \mathbf{h}_i^k \tag{51}$$

$$= \mathcal{T}^k(\mathbf{h}_i^k) + \mathbf{h}_i^k \tag{52}$$

$$\square$$

## H.2    HYPER-CONNECTION MATRIX OF PARALLEL ARRANGEMENT

In this section, we demonstrate that the following hyper-connection matrix will produce a network where every $n$ adjacent layers are arranged in parallel, with each layer incorporating residual connections. We define a parallel-arranged network such that $n$ adjacent layers form a group, with layers within a group being parallel and groups arranged sequentially. The output of $k$-th group is given by:

$$\mathbf{h}^{k+1} = \sum_{i=1}^{n} (\mathcal{T}^{k \times n + i}(\mathbf{h}^k) + \mathbf{h}^k).$$
(53)

It can be proved that this arrangement can be described by the following hyper-connection matrices.

**First, for $k$ where $k - 1 \equiv 0 \pmod{n}$:**

$$\mathcal{HC}^{\{k|k-1\equiv 0 \pmod{n}\}} = \begin{pmatrix} \mathbf{0}_{1\times 1} & \mathbf{e}_1^{\mathsf{T}} \\ \mathbf{1}_{n\times 1} & \mathbf{1}_{n\times n,} \end{pmatrix}$$
(54)

where the $\mathcal{HC}$ matrix can be decomposed into two operations: 1) sum up all the outputs of the previous group and use it as the input of the current layer and as the residual of the subsequent layers; 2) sum up the output and input saving to the first hidden vector slot.

**Next, for $k$ where $k - 1 \equiv i \pmod{n}$ and $i \neq 0$:**

$$\mathcal{HC}^{\{k|k-1\equiv i \pmod{n}, i\neq 0\}} = \begin{pmatrix} \mathbf{0}_{1\times 1} & \mathbf{e}_i^{\mathsf{T}} \\ \mathbf{e}_i & \mathbf{e}_{n\times n,} \end{pmatrix}.$$
(55)

where the $\mathcal{HC}$ matrix selects the $i$-th hidden vector as the input of the current layer, and sums up the output and input, saving to the $i$-th hidden vector slot.

This means:

$$\mathbf{h}^{k+1} = \mathcal{HC}^{(k+1)\times n}(\mathcal{T}^{(k+1)\times n},$$
(56)

$$\mathcal{HC}^{(k+1)\times n-1}(\mathcal{T}^{(k+1)\times n-1},$$
(57)

$$\cdots$$
(58)

$$\mathcal{HC}^{k\times n+1}(\mathcal{T}^{k\times n+1}, \mathbf{h}^k)))$$
(59)

This can also be proved by mathematical induction; however, the conclusion is quite obvious through drawing, and the proof process is very tedious. Therefore, we don't repeat the similar proof here.

# I  PSEUDOCODE OF HYPER-CONNECTIONS

---

**Algorithm 1** Network with Hyper-Connections

---

**Require:** Initial hidden vector $\mathbf{h}^0 \in \mathbb{R}^d$
**Require:** Expansion rate $n$
**Ensure:** Final output $\mathbf{y}$

1: **Initialize:**
2: $\mathbf{H}^0 \leftarrow \begin{pmatrix} \mathbf{h}^0 & \mathbf{h}^0 & \dots & \mathbf{h}^0 \end{pmatrix}^\mathsf{T} \in \mathbb{R}^{n \times d}$
3: **for** $k = 1$ to $L$ **do**                                        ▷ For each layer
4:     $\mathbf{H} \leftarrow \mathbf{H}^{k-1}$
5:     $(\mathbf{h}_0 \quad \mathbf{H}') \leftarrow \mathcal{WC}^{k\mathsf{T}}\mathbf{H}$              ▷ Width Connections
6:     $\mathbf{h}'_0 \leftarrow \mathcal{T}^k(\mathbf{h}_0)$                          ▷ Layer Computation
7:     $\hat{\mathbf{H}} \leftarrow \mathbf{B}^{k\mathsf{T}}\mathbf{h}'_0 + \mathbf{H}'$              ▷ Depth Connections
8:     $\mathbf{H}^k \leftarrow \hat{\mathbf{H}}$
9: **end for**
10: **Final Output:**
11: $\mathbf{h}^L \leftarrow$ sum rows of $\mathbf{H}^L$
12: $\mathbf{h}^L \leftarrow$ Normalization Layer($\mathbf{h}^L$)
13: $\mathbf{y} \leftarrow$ Output Layer($\mathbf{h}^L$)
14: **return** $\mathbf{y}$

---

## J PyTorch Implementation of Hyper-connections

---

**Algorithm 2** Pseudocode of hyper-connections in a PyTorch-like style.

---

```python
# h: hyper hidden matrix (BxLxNxD)

class HyperConnection(nn.Module):
    def __init__(self, dim, rate, layer_id, dynamic, device=None):
        super(HyperConnection, self).__init__()

        self.rate = rate
        self.layer_id = layer_id
        self.dynamic = dynamic

        self.static_beta = nn.Parameter(torch.ones((rate,), device=device))

        init_alpha0 = torch.zeros((rate, 1), device=device)
        init_alpha0[layer_id % rate, 0] = 1.
        self.static_alpha = nn.Parameter(torch.cat([init_alpha0, torch.eye((rate), device=
            device)], dim=1))

        if self.dynamic:
            self.dynamic_alpha_fn = nn.Parameter(torch.zeros((dim, rate+1), device=device))
            self.dynamic_alpha_scale = nn.Parameter(torch.ones(1, device=device) * 0.01)
            self.dynamic_beta_fn = nn.Parameter(torch.zeros((dim, ), device=device))
            self.dynamic_beta_scale = nn.Parameter(torch.ones(1, device=device) * 0.01)
            self.layer_norm = LayerNorm(dim)

    def width_connection(self, h):
        # get alpha and beta
        if self.dynamic:
            norm_h = self.layer_norm(h)

        if self.dynamic:
            wc_weight = norm_h @ self.dynamic_alpha_fn
            wc_weight = F.tanh(wc_weight)
            dynamic_alpha = wc_weight * self.dynamic_alpha_scale
            alpha = dynamic_alpha + self.static_alpha[None, None, ...]
        else:
            alpha = self.static_alpha[None, None, ...]

        if self.dynamic:
            dc_weight = norm_h @ self.dynamic_beta_fn
            dc_weight = F.tanh(dc_weight)
            dynamic_beta = dc_weight * self.dynamic_beta_scale
            beta = dynamic_beta + self.static_beta[None, None, ...]
        else:
            beta = self.static_beta[None, None, ...]

        # width connection
        mix_h = alpha.transpose(-1, -2) @ h

        return mix_h, beta

    def depth_connection(self, mix_h, h_o, beta):
        h = torch.einsum("blh,bln->blnh", h_o, beta) + mix_h[..., 1:, :]

        return h
```

---

**Algorithm 3** Pseudocode of transformer with hyper-connections in a PyTorch-like style.

---

```python
# h: hyper hidden matrix (BxLxNxD)
# atten_hyper_connection, ffn_hyper_connection: hyper-connection modules
# attn_norm, ffn_norm: normalization modules

# Attention Block
mix_h, beta = atten_hyper_connection.width_connection(h)
h = attn_norm(mix_h[...,0,:])
h = self_attention(h)
h = atten_hyper_connection.depth_connection(mix_h, dropout(h), beta)

# FFN Block
mix_h, beta = ffn_hyper_connection.width_connection(h)
h = ffn_norm(mix_h[...,0,:])
h = ffn(h)
h = ffn_hyper_connection.depth_connection(mix_h, dropout(h), beta)
```

---

## K    VALIDATION SETS AND DOWNSTREAM TASKS

Table 13: OLMo's default configuration was evaluated using multiple metrics. Perplexity (PPL) and loss were used for the V2 and V3 Validation Sets, while zero-shot testing was applied to the Downstream Benchmarks. However, the grey benchmarks were excluded from our analysis due to the instability of their performance indicators.

| **V2 Validation Sets** |
|---|
| `v2-small-4chan-validation` |
| `v2-small-c4_100_domains-validation` |
| `v2-small-c4_en-validation` |
| `v2-small-gab-validation` |
| `v2-small-ice-validation` |
| `v2-small-m2d2_s2orc-validation` |
| `v2-small-m2d2_wiki-validation` |
| `v2-small-manosphere-validation` |
| `v2-small-mc4_en-validation` |
| `v2-small-pile-validation` |
| `v2-small-ptb-validation` |
| `v2-small-twitterAEE-validation` |
| `v2-small-wikitext_103-validation` |
| **V3 Validation Sets** |
| `v3-small-c4_en-validation` |
| `v3-small-dolma_books-validation` |
| `v3-small-dolma_common-crawl-validation` |
| `v3-small-dolma_pes2o-validation` |
| `v3-small-dolma_reddit-validation` |
| `v3-small-dolma_stack-validation` |
| `v3-small-dolma_wiki-validation` |
| `v3-small-ice-validation` |
| `v3-small-m2d2_s2orc-validation` |
| `v3-small-pile-validation` |
| `v3-small-wikitext_103-validation` |
| **Downstream Benchmarks** |
| `piqa` (Bisk et al., 2020) |
| `hellaswag` (Zellers et al., 2019) |
| `winogrande` (Sakaguchi et al., 2021) |
| `openbook_qa` (Mihaylov et al., 2018) |
| `sciq` (Johannes Welbl, 2017) |
| `arc_easy` (Clark et al., 2018) |
| `copa` (Roemmele et al., 2011) |
| `commitment_bank` (De Marneffe et al., 2019) |
| `mrpc` (Dolan & Brockett, 2005) |
| `rte` (Dagan et al., 2005) |
| `sst2` (Socher et al., 2013) |

Table 14: Downstream Benchmarks for OLMoE.

| **Downstream Benchmarks for OLMoE** |
|---|
| `piqa` (Bisk et al., 2020) |
| `hellaswag` (Zellers et al., 2019) |
| `winogrande` (Sakaguchi et al., 2021) |
| `openbook_qa` (Mihaylov et al., 2018) |
| `sciq` (Johannes Welbl, 2017) |
| `arc_easy` (Clark et al., 2018) |
| `arc_challenage` (Clark et al., 2018) |
| `copa` (Roemmele et al., 2011) |
| `boolq` (Clark et al., 2019) |
| `commonsense_qa` (Talmor et al., 2018) |
| `social_iqa` (Sap et al., 2019) |
| `mmlu` (Hendrycks et al., 2021) |

## L  1B MODEL EXPERIMENTS

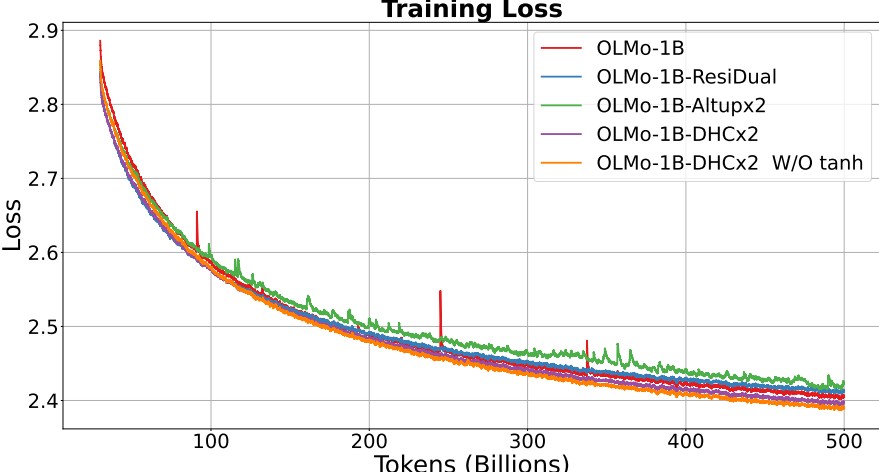

Figure 15: Training loss curves of related works, smoothed using Exponential Moving Average (EMA) with a decay rate of 0.99.

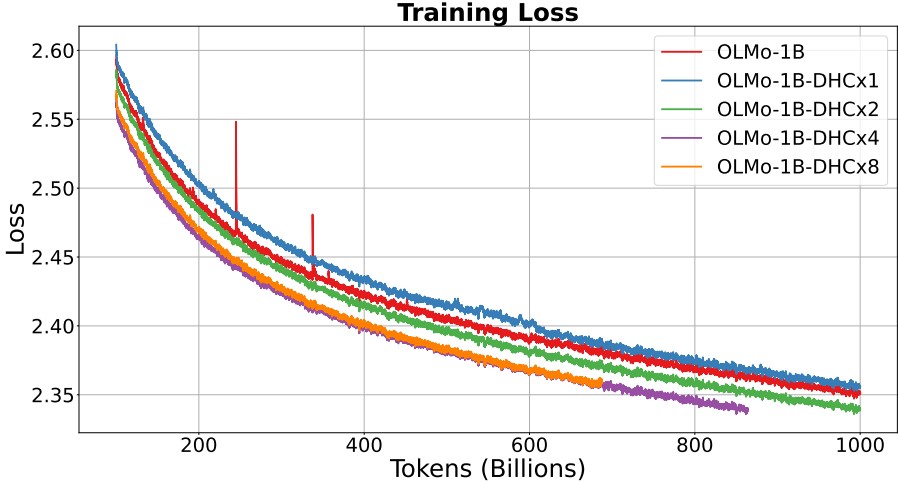

Figure 16: Training loss curves of DHC with `tanh` over 500 billion tokens, smoothed using Exponential Moving Average (EMA) with a decay rate of 0.99.

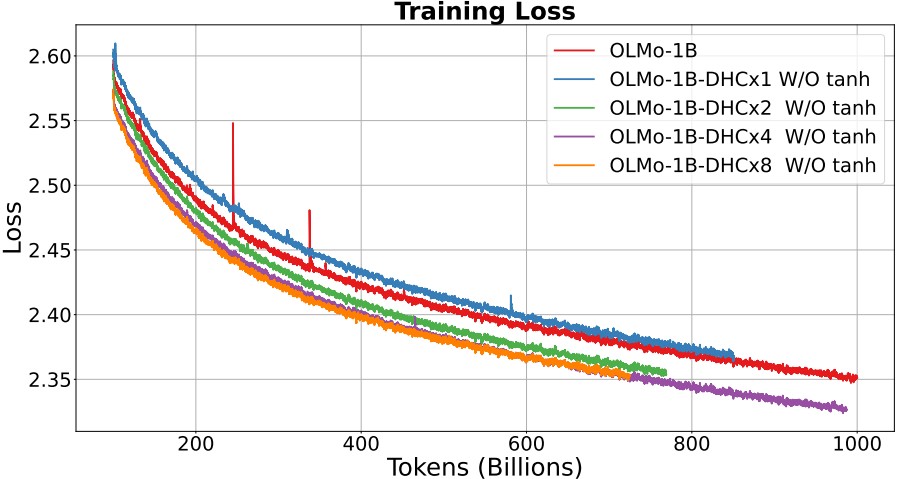

Figure 17: Training loss curves of DHC without `tanh` over 500 billion tokens, smoothed using Exponential Moving Average (EMA) with a decay rate of 0.99.

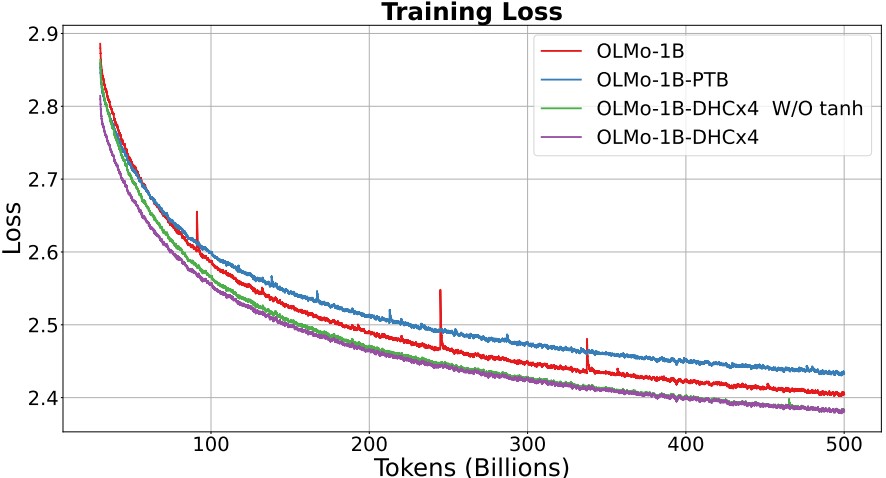

Figure 18: Training loss curves compared with parallel transformer blocks (PTB), smoothed using Exponential Moving Average (EMA) with a decay rate of 0.99.

Table 15: Results on downstream benchmarks for 1B models.

| Method | arc_easy | copa | hellaswag | openbook_qa | piqa | sciq | winogrande | avg. |
|--------|----------|------|-----------|-------------|------|------|------------|------|
| OLMo-1B | 56.8 | 76.0 | 56.1 | 33.8 | 74.4 | 85.1 | 55.6 | 62.5 |
| *Scaling n in DHC W/O tanh* | | | | | | | | |
| OLMo-1B-DHCx1 W/O tanh | 56.8 | 75.0 | 55.3 | 33.4 | 72.9 | 85.4 | 57.1 | 62.3 |
| OLMo-1B-DHCx2 W/O tanh | 63.0 | 74.0 | 57.1 | 34.6 | 73.5 | 86.0 | 58.2 | 63.8 |
| OLMo-1B-DHCx4 W/O tanh | 61.2 | 80.0 | 57.5 | 33.6 | 75.5 | 85.8 | 56.9 | 64.4 |
| OLMo-1B-DHCx8 W/O tanh | 61.1 | 75.0 | 57.6 | 35.4 | 73.8 | 85.2 | 58.5 | 63.8 |
| *Scaling n in DHC* | | | | | | | | |
| OLMo-1B-DHCx1 | 59.7 | 74.0 | 55.5 | 33.6 | 73.5 | 85.4 | 54.5 | 62.3 |
| OLMo-1B-DHCx2 | 59.7 | 73.0 | 56.7 | 34.0 | 74.7 | 85.2 | 57.9 | 63.0 |
| OLMo-1B-DHCx4 | 59.8 | 79.0 | 58.1 | 32.4 | 74.3 | 86.1 | 57.1 | 63.8 |
| OLMo-1B-DHCx8 | 56.8 | 75.0 | 58.0 | 34.4 | 73.8 | 84.2 | 57.3 | 62.8 |
| *Scaling n in SHC* | | | | | | | | |
| OLMo-1B-SHCx2 | 59.1 | 77.0 | 56.6 | 35.4 | 74.2 | 85.3 | 56.4 | 63.4 |
| OLMo-1B-SHCx4 | 59.3 | 77.0 | 56.7 | 34.0 | 74.3 | 86.6 | 57.1 | 63.6 |
| *Non-trainable $\mathcal{WC}$* | | | | | | | | |
| OLMo-1B-DHCx4 | 60.5 | 78.0 | 56.2 | 34.0 | 73.5 | 86.0 | 55.8 | 63.4 |
| OLMo-1B-DHCx4 W/O tanh | 59.1 | 72.0 | 56.8 | 35.0 | 73.3 | 86.0 | 55.5 | 62.5 |
| *Non-trainable $\mathbf{B}$* | | | | | | | | |
| OLMo-1B-DHCx4 | 59.5 | 77.0 | 57.9 | 33.8 | 73.3 | 85.6 | 56.6 | 63.4 |
| OLMo-1B-DHCx4 W/O tanh | 60.4 | 74.0 | 57.6 | 34.0 | 74.9 | 86.7 | 57.5 | 63.6 |

Table 16: Losses of V2 validation sets for 1B Model.

| Method | 4chan | c4_100_domains | c4_en | gab | ice | m2d2_s2orc | m2d2_wiki | manosphere | mc4_en | pile | ptb | twitterAAE | wikitext_103 | avg |
|---|---|---|---|---|---|---|---|---|---|---|---|---|---|---|
| OLMo-1B | 2.319 | 2.615 | 2.762 | 3.364 | 2.719 | 3.085 | 2.594 | 3.028 | 2.522 | 2.250 | 2.953 | 3.672 | 2.657 | 2.811 |
| Scaling n in DHC W/O tanh | | | | | | | | | | | | | | |
| OLMo-1B-DHCx1 W/O tanh | 2.320 | 2.626 | 2.773 | 3.379 | 2.725 | 3.102 | 2.609 | 3.036 | 2.531 | 2.264 | 2.948 | 3.703 | 2.672 | 2.822 |
| OLMo-1B-DHCx2 W/O tanh | 2.311 | 2.600 | 2.749 | 3.362 | 2.700 | 3.069 | 2.583 | 3.015 | 2.503 | 2.231 | 2.908 | 3.635 | 2.625 | 2.792 |
| OLMo-1B-DHCx4 W/O tanh | 2.295 | 2.591 | 2.735 | 3.344 | 2.686 | 3.056 | 2.562 | 3.005 | 2.492 | 2.221 | 2.898 | 3.632 | 2.610 | 2.779 |
| OLMo-1B-DHCx8 W/O tanh | 2.292 | 2.589 | 2.734 | 3.350 | 2.685 | 3.060 | 2.562 | 3.006 | 2.492 | 2.218 | 2.878 | 3.628 | 2.609 | 2.777 |
| Scaling n in DHC | | | | | | | | | | | | | | |
| OLMo-1B-DHCx1 | 2.323 | 2.625 | 2.775 | 3.376 | 2.728 | 3.090 | 2.606 | 3.037 | 2.533 | 2.262 | 2.961 | 3.652 | 2.678 | 2.819 |
| OLMo-1B-DHCx2 | 2.309 | 2.608 | 2.754 | 3.367 | 2.703 | 3.061 | 2.587 | 3.022 | 2.509 | 2.237 | 2.930 | 3.704 | 2.636 | 2.802 |
| OLMo-1B-DHCx4 | 2.290 | 2.591 | 2.738 | 3.354 | 2.683 | 3.064 | 2.564 | 3.005 | 2.492 | 2.218 | 2.890 | 3.641 | 2.611 | 2.781 |
| OLMo-1B-DHCx8 | 2.295 | 2.591 | 2.739 | 3.353 | 2.684 | 3.054 | 2.567 | 3.008 | 2.493 | 2.219 | 2.876 | 3.631 | 2.608 | 2.778 |
| Scaling n in SHC | | | | | | | | | | | | | | |
| OLMo-1B-SHCx2 | 2.307 | 2.610 | 2.757 | 3.360 | 2.703 | 3.063 | 2.587 | 3.023 | 2.511 | 2.238 | 2.933 | 3.643 | 2.643 | 2.799 |
| OLMo-1B-SHCx4 | 2.300 | 2.603 | 2.751 | 3.357 | 2.692 | 3.062 | 2.580 | 3.018 | 2.504 | 2.232 | 2.899 | 3.653 | 2.627 | 2.791 |
| Non-trainable WC | | | | | | | | | | | | | | |
| OLMo-1B-DHCx4 | 2.312 | 2.608 | 2.752 | 3.357 | 2.700 | 3.077 | 2.583 | 3.024 | 2.508 | 2.238 | 2.959 | 3.678 | 2.636 | 2.802 |
| OLMo-1B-DHCx4 W/O tanh | 2.308 | 2.609 | 2.755 | 3.357 | 2.710 | 3.100 | 2.585 | 3.025 | 2.510 | 2.240 | 2.945 | 3.663 | 2.644 | 2.804 |
| Non-trainable Beta | | | | | | | | | | | | | | |
| OLMo-1B-DHCx4 | 2.296 | 2.594 | 2.742 | 3.348 | 2.684 | 3.051 | 2.569 | 3.008 | 2.497 | 2.221 | 2.917 | 3.627 | 2.622 | 2.783 |
| OLMo-1B-DHCx4 W/O tanh | 2.295 | 2.592 | 2.739 | 3.347 | 2.689 | 3.066 | 2.567 | 3.005 | 2.496 | 2.222 | 2.887 | 3.638 | 2.606 | 2.781 |

Table 17: Perplexities of V2 validation sets for 1B models.

| Method | 4chan | c4_100_domains | c4_en | gab | ice | m2d2_s2orc | m2d2_wiki | manosphere | mc4_en | pile | ptb | twitterAAE | wikitext_103 | avg |
|---|---|---|---|---|---|---|---|---|---|---|---|---|---|---|
| OLMo-1B | 10.167 | 13.666 | 15.829 | 28.901 | 15.166 | 21.860 | 13.377 | 20.651 | 12.453 | 9.488 | 19.161 | 39.328 | 14.251 | 18.023 |
| Scaling n in DHC W/O tanh | | | | | | | | | | | | | | |
| OLMo-1B-DHCx1 W/O tanh | 10.174 | 13.815 | 16.004 | 29.328 | 15.259 | 22.231 | 13.587 | 20.823 | 12.562 | 9.620 | 19.071 | 40.580 | 14.462 | 18.270 |
| OLMo-1B-DHCx2 W/O tanh | 9.920 | 13.340 | 15.412 | 28.340 | 14.676 | 21.243 | 12.965 | 20.181 | 12.079 | 9.219 | 18.129 | 37.768 | 13.594 | 17.451 |
| OLMo-1B-DHCx4 W/O tanh | 10.082 | 13.470 | 15.625 | 28.848 | 14.882 | 21.521 | 13.234 | 20.392 | 12.217 | 9.312 | 18.321 | 37.905 | 13.806 | 17.663 |
| OLMo-1B-DHCx8 W/O tanh | 9.897 | 13.313 | 15.387 | 28.488 | 14.658 | 21.337 | 12.960 | 20.200 | 12.084 | 9.185 | 17.782 | 37.650 | 13.592 | 17.425 |
| Scaling n in DHC | | | | | | | | | | | | | | |
| OLMo-1B-DHCx1 | 10.210 | 13.810 | 16.031 | 29.265 | 15.302 | 21.986 | 13.539 | 20.847 | 12.584 | 9.606 | 19.326 | 38.564 | 14.555 | 18.125 |
| OLMo-1B-DHCx2 | 10.061 | 13.568 | 15.710 | 29.002 | 14.925 | 21.349 | 13.284 | 20.524 | 12.294 | 9.362 | 18.727 | 40.592 | 13.957 | 17.950 |
| OLMo-1B-DHCx4 | 9.877 | 13.344 | 15.430 | 28.624 | 14.633 | 21.410 | 13.006 | 20.186 | 12.080 | 9.189 | 18.102 | 38.136 | 13.606 | 17.509 |
| OLMo-1B-DHCx8 | 9.922 | 13.346 | 15.467 | 28.591 | 14.640 | 21.198 | 13.025 | 20.240 | 12.097 | 9.196 | 17.749 | 37.743 | 13.570 | 17.445 |
| Scaling n in SHC | | | | | | | | | | | | | | |
| OLMo-1B-SHCx2 | 10.046 | 13.601 | 15.753 | 28.782 | 14.931 | 21.391 | 13.294 | 20.562 | 12.319 | 9.374 | 18.791 | 38.212 | 14.060 | 17.778 |
| OLMo-1B-SHCx4 | 9.977 | 13.507 | 15.655 | 28.691 | 14.766 | 21.372 | 13.194 | 20.457 | 12.234 | 9.315 | 18.149 | 38.569 | 13.836 | 17.671 |
| Non-trainable WC | | | | | | | | | | | | | | |
| OLMo-1B-DHCx4 | 10.054 | 13.587 | 15.721 | 28.689 | 15.023 | 22.186 | 13.263 | 20.594 | 12.310 | 9.390 | 19.016 | 38.959 | 14.070 | 17.912 |
| OLMo-1B-DHCx4 W/O tanh | 10.092 | 13.566 | 15.666 | 28.704 | 14.873 | 21.696 | 13.242 | 20.579 | 12.276 | 9.377 | 19.272 | 39.570 | 13.963 | 17.914 |
| Non-trainable Beta | | | | | | | | | | | | | | |
| OLMo-1B-DHCx4 | 9.927 | 13.354 | 15.475 | 28.417 | 14.722 | 21.454 | 13.021 | 20.185 | 12.135 | 9.228 | 17.932 | 38.005 | 13.553 | 17.493 |
| OLMo-1B-DHCx4 W/O tanh | 9.932 | 13.386 | 15.510 | 28.436 | 14.641 | 21.130 | 13.051 | 20.253 | 12.142 | 9.220 | 18.478 | 37.610 | 13.766 | 17.504 |

Table 18: Losses of V3 validation sets for 1B model.

| Method | c4_en | dolma_books | dolma_common-crawl | dolma_pes2o | dolma_reddit | dolma_stack | dolma_wiki | ice | m2d2_s2orc | pile | wikitext_103 | avg |
|---|---|---|---|---|---|---|---|---|---|---|---|---|
| OLMo-1B | 2.702 | 2.906 | 2.722 | 2.333 | 2.980 | 1.041 | 2.487 | 2.715 | 3.199 | 2.232 | 2.663 | 2.544 |
| *Scaling n in DHC W/O tanh* | | | | | | | | | | | | |
| OLMo-1B-DHCx1 W/O tanh | 2.712 | 2.928 | 2.732 | 2.349 | 2.991 | 1.045 | 2.499 | 2.721 | 3.219 | 2.246 | 2.677 | 2.556 |
| OLMo-1B-DHCx2 W/O tanh | 2.676 | 2.880 | 2.698 | 2.306 | 2.961 | 1.024 | 2.456 | 2.682 | 3.174 | 2.204 | 2.617 | 2.516 |
| OLMo-1B-DHCx4 W/O tanh | 2.689 | 2.890 | 2.706 | 2.317 | 2.969 | 1.030 | 2.471 | 2.697 | 3.200 | 2.213 | 2.633 | 2.529 |
| OLMo-1B-DHCx8 W/O tanh | 2.674 | 2.876 | 2.695 | 2.303 | 2.960 | 1.022 | 2.454 | 2.680 | 3.176 | 2.200 | 2.616 | 2.514 |
| *Scaling n in DHC* | | | | | | | | | | | | |
| OLMo-1B-DHCx1 | 2.714 | 2.927 | 2.732 | 2.346 | 2.991 | 1.045 | 2.499 | 2.723 | 3.211 | 2.245 | 2.683 | 2.556 |
| OLMo-1B-DHCx2 | 2.694 | 2.901 | 2.712 | 2.321 | 2.976 | 1.032 | 2.478 | 2.699 | 3.202 | 2.218 | 2.642 | 2.534 |
| OLMo-1B-DHCx4 | 2.675 | 2.876 | 2.697 | 2.301 | 2.962 | 1.021 | 2.455 | 2.679 | 3.176 | 2.200 | 2.617 | 2.515 |
| OLMo-1B-DHCx8 | 2.677 | 2.880 | 2.701 | 2.304 | 2.964 | 1.022 | 2.456 | 2.680 | 3.177 | 2.201 | 2.614 | 2.516 |
| *Scaling n in SHC* | | | | | | | | | | | | |
| OLMo-1B-SHCx2 | 2.698 | 2.907 | 2.718 | 2.325 | 2.980 | 1.032 | 2.479 | 2.700 | 3.198 | 2.221 | 2.650 | 2.537 |
| OLMo-1B-SHCx4 | 2.689 | 2.892 | 2.711 | 2.315 | 2.973 | 1.028 | 2.472 | 2.688 | 3.195 | 2.214 | 2.633 | 2.528 |
| *Non-trainable WC* | | | | | | | | | | | | |
| OLMo-1B-DHCx4 | 2.695 | 2.903 | 2.716 | 2.324 | 2.978 | 1.035 | 2.477 | 2.705 | 3.201 | 2.221 | 2.649 | 2.537 |
| OLMo-1B-DHCx4 W/O tanh | 2.692 | 2.899 | 2.714 | 2.321 | 2.976 | 1.032 | 2.474 | 2.695 | 3.189 | 2.219 | 2.641 | 2.532 |
| *Non-trainable Beta* | | | | | | | | | | | | |
| OLMo-1B-DHCx4 | 2.679 | 2.880 | 2.697 | 2.306 | 2.961 | 1.025 | 2.458 | 2.684 | 3.188 | 2.204 | 2.612 | 2.518 |
| OLMo-1B-DHCx4 W/O tanh | 2.681 | 2.886 | 2.702 | 2.306 | 2.966 | 1.024 | 2.462 | 2.680 | 3.183 | 2.204 | 2.628 | 2.520 |

Table 19: Perplexities of V3 validation sets for 1B models.

| Method | c4_en | dolma_books | dolma_common-crawl | dolma_pes2o | dolma_reddit | dolma_stack | dolma_wiki | ice | m2d2_s2orc | pile | wikitext_103 | avg |
|---|---|---|---|---|---|---|---|---|---|---|---|---|
| OLMo-1B | 14.908 | 18.289 | 15.216 | 10.305 | 19.686 | 2.832 | 12.026 | 15.098 | 24.503 | 9.319 | 14.334 | 14.229 |
| *Scaling n in DHC W/O tanh* | | | | | | | | | | | | |
| OLMo-1B-DHCx1 W/O tanh | 15.064 | 18.699 | 15.356 | 10.473 | 19.909 | 2.843 | 12.167 | 15.191 | 25.013 | 9.451 | 14.540 | 14.428 |
| OLMo-1B-DHCx2 W/O tanh | 14.531 | 17.817 | 14.857 | 10.038 | 19.323 | 2.783 | 11.662 | 14.608 | 23.906 | 9.061 | 13.694 | 13.844 |
| OLMo-1B-DHCx4 W/O tanh | 14.711 | 17.996 | 14.975 | 10.146 | 19.479 | 2.800 | 11.830 | 14.839 | 24.524 | 9.146 | 13.917 | 14.033 |
| OLMo-1B-DHCx8 W/O tanh | 14.494 | 17.749 | 14.813 | 10.000 | 19.306 | 2.779 | 11.630 | 14.587 | 23.948 | 9.021 | 13.684 | 13.819 |
| *Scaling n in DHC* | | | | | | | | | | | | |
| OLMo-1B-DHCx1 | 15.093 | 18.675 | 15.360 | 10.442 | 19.909 | 2.845 | 12.174 | 15.225 | 24.810 | 9.436 | 14.632 | 14.418 |
| OLMo-1B-DHCx2 | 14.794 | 18.190 | 15.061 | 10.191 | 19.612 | 2.806 | 11.915 | 14.870 | 24.589 | 9.187 | 14.043 | 14.114 |
| OLMo-1B-DHCx4 | 14.514 | 17.743 | 14.829 | 9.989 | 19.343 | 2.776 | 11.650 | 14.573 | 23.948 | 9.028 | 13.689 | 13.826 |
| OLMo-1B-DHCx8 | 14.546 | 17.807 | 14.889 | 10.011 | 19.366 | 2.779 | 11.653 | 14.579 | 23.964 | 9.030 | 13.653 | 13.843 |
| *Scaling n in SHC* | | | | | | | | | | | | |
| OLMo-1B-SHCx2 | 14.854 | 18.293 | 15.150 | 10.230 | 19.689 | 2.807 | 11.934 | 14.876 | 24.478 | 9.214 | 14.150 | 14.152 |
| OLMo-1B-SHCx4 | 14.717 | 18.028 | 15.049 | 10.121 | 19.550 | 2.796 | 11.846 | 14.699 | 24.407 | 9.155 | 13.912 | 14.025 |
| *Non-trainable WC* | | | | | | | | | | | | |
| OLMo-1B-DHCx4 | 14.810 | 18.224 | 15.120 | 10.215 | 19.650 | 2.816 | 11.902 | 14.954 | 24.552 | 9.220 | 14.135 | 14.145 |
| OLMo-1B-DHCx4 W/O tanh | 14.756 | 18.160 | 15.095 | 10.191 | 19.613 | 2.806 | 11.868 | 14.807 | 24.273 | 9.203 | 14.021 | 14.072 |
| *Non-trainable Beta* | | | | | | | | | | | | |
| OLMo-1B-DHCx4 | 14.574 | 17.820 | 14.840 | 10.038 | 19.320 | 2.787 | 11.677 | 14.647 | 24.233 | 9.059 | 13.621 | 13.874 |
| OLMo-1B-DHCx4 W/O tanh | 14.593 | 17.926 | 14.904 | 10.032 | 19.405 | 2.785 | 11.724 | 14.588 | 24.108 | 9.060 | 13.839 | 13.906 |

