# OpenReview forum: "Hyper-Connections"
_ICLR.cc/2025/Conference — ICLR 2025 Poster_

### Official Review · Reviewer_fvae · 2024-11-03

**Soundness:** 2
**Presentation:** 2
**Contribution:** 2
**Rating:** 5
**Confidence:** 5

**Summary:**

The paper introduces hyper-connections, a approach that aims to address the limitations of residual connections of transformers architectures.
The hyper-connections approach introduce depth-connections and width-connections, allowing a more customizable interaction between layers. Depth-connections are weighted connections across different layers, while width-connections facilitate two layers at the same depths. More importantly the hyper-connections are learned, so it improves model adaptability.
Models with hyper-connections have significant performance improvements over the original architectures.

**Strengths:**

The experimental results seems good enough to claim the hyper-connections is better than the original architectures.

**Weaknesses:**

1. The presentation is horrible, it is really hard to understand what is model is doing. Probably a better figure or even pseudocode should be provided to explain the methods.
2. Although the experimental results are good, there is no information on how much extra computation is need to achieve such good results. I suspected that the extra performance is highly due to the extra parameters or extra computation it needed. If you make your model have the same floats, probably you could see that the performance is really similar to the original transformers.
3. The ordering of the paper is horrible, there is almost no explanation why you do each thing in the paper.

**Questions:**

1. In the introduction, you stated that "Post-Norm applies normalization operations after the output of each residual block, weakening the
"strength" of residuals." I don't think it is correct, since post-norm applies normalization after the summation of the skip and residual branch, so it shouldn't weaken the strength of the residuals.

2. Can you also should the flops count for your model and the original model?

---

> ### Author Response · Authors · 2024-11-21
>
> We greatly appreciate the time you have taken to review our manuscript. In response to your comments, we address each point individually.
>
> **Weakness1:**
> Thank you for your suggestions. We have revised Figure 2 and added an additional architecture diagram in Figure 8 of Appendix A for more intuitive illustration. You can also check the special $ n = 2 $ case in Figure 4 for more clear understanding. For the pseudocode, see Algorithm 1.
> The hyper-connection is designed to address the gradient vanishing and representation collapse challenges stemming from Pre-Norm and Post-Norm. Our motivation is to enable neural networks to autonomously learn the optimal strength of connections to improve performance. Unlike typical residual connections that only connect the input vector and the layer output vector, we expand the layer input to $ n $ input vectors, connect different input vectors (width-connections), feed them into the layer to get the output vector, and then further connect the input vectors and the layer output vector (depth-connections).
>
> We greatly appreciate specific suggestions for enhancement. Reviewer VJDJ praised the paper as “well detailed and mathematically sound,” DUBh noted “thorough analysis” and “nice visualizations,” and HHYn commended it as “a clear and systematic extension”. We kindly request that if you encountered any sections that were difficult to understand or felt could be improved, you could point these out specifically. This feedback would be invaluable in enhancing the readability and clarity of our paper.
>
> **Weakness2:**
> To clarify, the number of parameters and the computational cost of our model are almost the same as the original model, with the additional parameters and computation being at most a fraction of a thousandth compared to the original model.
> We have added a table of parameter counts and computational costs in Appendix B. Since there is still space in the table for the 7B experiments, we have added the parameters and computation (FLOPs) to the 7B experiment table.
> The detailed data is presented as follows.
>
> ### Table: Comparison of number of parameters
>
> | **Method**          | **HC Params (B)** | **Total Params (B)** | **Total Params Δ rate (%)** |
> |---------------------|-------------------|----------------------|-----------------------------|
> | **OLMo-1B**         | -                 | 1.17676442           | -                           |
> | **OLMo-1B-SHCx2**   | 0.0000026         | 1.17676467           | **+0.00002%**               |
> | **OLMo-1B-SHCx4**   | 0.0000077         | 1.17676518           | **+0.00007%**               |
> | **OLMo-1B-DHCx2**   | 0.0002625         | 1.17702688           | **+0.02230%**               |
> | **OLMo-1B-DHCx4**   | 0.0003940         | 1.17715846           | **+0.03349%**               |
> | **OLMo-7B**         | -                 | 6.88809574           | -                           |
> | **OLMo-7B-DHCx4**   | 0.0013124         | 6.88967027           | **+0.02286%**               |
> | **OLMoE-1B-7B**     | -                 | 6.91909427           | -                           |
> | **OLMoE-1B-7B-DHCx4** | 0.0003940       | 6.91948832           | **+0.00570%**               |
> ### Table: FLOPs per token in forward pass
>
> | **Method**            | **HC FLOPs (G)** | **Total FLOPs (G)** | **Total FLOPs Δ rate (%)** |
> |-----------------------|------------------|---------------------|----------------------------|
> | **OLMo-1B**           | -                | 2.3536              | -                          |
> | **OLMo-1B-SHCx2**     | 0.0010           | 2.3545              | **+0.038%**                 |
> | **OLMo-1B-SHCx4**     | 0.0031           | 2.3566              | **+0.127%**                 |
> | **OLMo-1B-DHCx2**     | 0.0020           | 2.3554              | **+0.076%**                 |
> | **OLMo-1B-DHCx4**     | 0.0049           | 2.3583              | **+0.200%**                 |
> | **OLMo-7B**       | -                | 13.3647             | -                          |
> | **OLMoE-7B-DHCx4** | 0.0197           | 13.3844             | **+0.147%**                 |
> | **OLMoE-1B-7B**       | -                | 2.3580              | -                          |
> | **OLMoE-1B-7B-DHCx4** | 0.0049           | 2.3629              | **+0.208%**                 |

---

> ### Author Response · Authors · 2024-11-21
>
> **Weakness3:**
> Sorry for any confusion regarding the structure and flow of our paper. Let us provide a clearer overview of the organization and the rationale behind each section:
>
> ### Section 1: Introduction
>
> (1) We begin by outlining our motivation: enabling neural networks to autonomously learn the optimal strength of connections to improve performance.
>
> (2) We introduce our key idea: expanding the layer input to $n$ input vectors, connecting different input vectors (width-connections), feeding them into the layer to get the output vector, and further connecting the input vectors and the layer output vector (depth-connections).
>
> ### Section 2: Method
>
> 2.1: We formally define the mathematical formulation of $\mathcal{HC}$ for controlling depth-connections and width-connections mentioned in Section 1
>
> 2.2: We present a dynamic version where $\mathcal{HC}$ depends on the input, achieving even better performance.
>
> 2.3: We explain the initialization for $\mathcal{HC}$, which is crucial for training convergence.
>
> ### Section 3: Further Analysis
>
> 3.1: We compare our approach with ordinary PostNorm/PreNorm, demonstrating that they are special cases of our hyper-connections, thereby making our method broadly applicable.
>
> 3.2: We discuss sequential and parallel duality, showing that hyper-connections can learn to arrange layers, which is a promising direction for designing more representative foundation models.
>
> ### Section 4: Experiments
>
> (1)We present comprehensive ablation studies and comparisons with OLMo and OLMoE models using 6 tables and 6 figures.
>
> (2)We include visualization analysis to demonstrate that our model learns parallel block patterns and highlights some interesting findings.
>
> ### Section 5 and Section 6
>
> We review relevant literature and summarize our contributions
>
> We greatly appreciate the positive feedback from reviewers VJDJ, DUBh, and HHYn, who praised the paper for being “a clear and systematic extension”. We are committed to ensuring that our paper is as clear and well-structured as possible. If you have any specific sections that you find difficult to understand or feel could be improved, please let us know. Your feedback will be invaluable in enhancing the readability and clarity of our paper.
>
> **Question1:**
> We sincerely appreciate the feedback and recognize that our original explanation may not have been sufficiently clear.
> In the paper, we rephrased it as follows:
>  "In contrast, Post-Norm applies normalization after the output of each residual block, reducing the influence of a hidden state on subsequent layers."
>
> What we intended to convey is that the influence of the output hidden state of each layer on the subsequent layers decreases. Specifically, suppose $h_i$ is the output of the i-th layer, and $h_j$ is the output of the j-th layer, where i>j.
>
> For PreNorm, we have:
> $h_i = L_{i-1}(h_{i-1}) + h_{i-1} = L_{i-1}(h_{i-1}) + L_{i-2}(h_{i-2}) + h_{i-2} = \Sigma_{k=j}^{i-1} L_k(h_k) + h_j$
>
> For PostNorm, we have:$h_{i}=\texttt{Norm}(L(h_{i-1})+h_{i-1})
> =\frac{L(h_{i-1})+h_{i-1}}{\sqrt{var(L(h_{i-1}))+var(h_{i-1})+2\cdot covar(L(h_{i-1}), h_{i-1}))}}$
>
> Let:
> $w_{i-1}=\frac{L(h_{i-1})+h_{i-1}}{\sqrt{var(L(h_{i-1}))+var(h_{i-1})+2\cdot covar(L(h_{i-1}), h_{i-1}))}}$,
>
> Typically, we assume $covar(L(h_{i}), h_{i}))=0$
> , as stated in [Section 4.1 of the paper](https://arxiv.org/pdf/2004.08249).
>
> Since $h_{i}$ has already been normalized, $var(h_{i})=1$, and therefore $w_{i-1}<1$.
> Finally, we have:
> $h_i=w_{i-1}L_{i-1}(h_{i-1}) + w_{i-1}h_{i-1}
> =w_{i-1}L_{i-1}(h_{i-1}) + w_{i-1}(w_{i-2}L_{i-1}(h_{i-2}) + w_{i-2}h_{i-2})=\Sigma_{k=j}^{i-1}(\Pi_{a=k}^{i-1}w_a)L_k(h_k)+\Pi_{k=j}^{i-1}w_kh_j.$
>
> $\Pi_{k=j}^{i-1}w_k<1$ represents the influence factor of $h_j$ on $h_i$. Since the product of several values less than 1 decays rapidly, the influence of $h_j$ on subsequent outputs diminishes as the number of layers increases.
>
> **Question2:** Yes, we have added comparisons of the parameter count and FLOPs, as well as the formulas for calculating the parameter count, in Appendix B. Additionally, we have included the table in response to Weakness 2.

---

> ### Author Response · Authors · 2024-11-23
>
> We sincerely appreciate your thoughtful and constructive feedback. Your insights are invaluable in improving our work, and we remain open to further discussion should you have any additional questions or concerns about our response.
>
> As all reviewers have recognized, our results are highly promising while adding virtually no extra computational cost. We believe our approach will become increasingly practical and significant in the era of large language models. We hope these insights and outcomes will contribute meaningfully to the community. We truly appreciate your time and would be very grateful if you could re-evaluate the paper’s rating.

---

> ### Author Response · Authors · 2024-11-26
>
> **Supplementary Response to Weakness1:** Regarding the pseudocode, actually, we have provided it in Appendix G and H of the original version. In the new version, these pseudocodes can be found in Appendix I and J.

---

> ### Comment · Reviewer_fvae · 2024-11-27
>
> Given the author does address most of my concerns raised in the original review, I will raise my rating to 5 to reflect the new manuscript.

---

> ### Author Response · Authors · 2024-11-29
>
> Thank you for taking the time to reevaluate our work. We noticed that your rating remains on the negative side. We are concerned that there may be additional points we have not fully aligned with or other issues that remain unresolved. If so, we would be more than happy to provide further clarification or address these concerns.

---

### Official Review · Reviewer_VJDJ · 2024-11-03

**Soundness:** 4
**Presentation:** 3
**Contribution:** 3
**Rating:** 8
**Confidence:** 3

**Summary:**

The paper introduced hyperconnection - width and depth connections with learnable strengths. Hyperconnections are proposed to find a balance between the seesaw effect noticed between vanishing gradients and representation collapse. The input is split into n copies for different connections, and these different copies are added together before layer computation.  The paper also introduced dynamic hyperconnections that dynamically update the strength of connections based on inputs. The paper performs comprehensive analysis of effect of hyperconnections on OLMO and OLMOE and image generation and classifcation problems and investigates its effect with prominent residual connections.

**Strengths:**

1. The results on LLM benchmarks and losses suggest a better balance between vanishing gradients and representation collapse.

2. Section 4.5 discusses the effect of hyperconnections, which displays that hyperconnections eliminate input embeddings from the output, form parallel blocks which have less reliance on each other increasing chances for unique representations.

3.Parallel block formation is particularly important as similar layers in a transformer block tends to learn similar representations. The ability to parallely form blocks dynamically based on input reduces the chances for similar representations enabling better models.

4. Assessing the hyperconnections can reveal some internal logic of the neural network. For instance, we may find that a set of classes follow similar paths compared to other classes.

5. The paper is well detailed and mathematically sound, proofs, hyperparameters and other implementation details are discussed appropriately.

**Weaknesses:**

1. The main drawback is when creating $n$ copies,  it leads to a considerable amount of increase in memory, though the burden can be reduced through engineering, the impact is yet to be known.

2. If the goal of creating multiple copies is just to make sure multiple depth connections can be modelled parallelly, is creating such copies actually necessary? Can't a single copy be used with different residual strengths? The only difference would be in gradient computation , were additional terms for each depth connection would be added to the singular copy. Can an ablation be conducted between DHC ($n=4$ and $n=1$) and SHC ($n=4$ an $n=1$) to show the importance and need for additional copies. What I am particularly interested is the advantages with updating the gradients into different copies and then adding them before every layer computation verses  updating all the gradients with a single copy. If there is any other specific need for expansion please free to refute this point.

3. Figure 2 caption improvement. The figure can discuss the diagram better, whats $\beta$, and need to add $\alpha_{1,0}$ and $\alpha_{2,0}$ be explained in the diagram itself. This is important as Figure 2 is the central figure that tries to encompass hyperconnections therefore adding these information would make it more clearer for readers.

**Questions:**

Some questions that may add more value to the paper:
1. With Dynamic Hyperconnections, there is a possibility for redundant connections (when strength becomes 0) how does these cases affect the seesaw effect of gradient vanishing and representation collapse.
2. Do DHC connections behave similar for images belonging to the same class in case of image classification?

Things I expect from the rebuttal are actions on points 2 and 3.

---

> ### Comment · Reviewer_VJDJ · 2024-11-21
> **Post Rebuttal Comments**
>
> I read the rebuttal and respective updations in the paper, I am mostly satisfied with the rebuttal. My rating remains unchanged.

---

> ### Author Response · Authors · 2024-11-21
>
> We greatly appreciate the time and effort you have taken to review our manuscript. In response to your insightful comments, we address each point individually.
>
> **Weakness1:**
> Sorry for the confusion. To clarify, the hidden state is duplicated into $ n $ copies only once at the beginning of the network input. Then, each layer of our hyper-connections accepts $ n $ hidden vector inputs, feeds them into the transformer layer, and applies residual connections with reweighting, controlled by the $\mathcal{HC}$ matrix of $\mathbb{R}^{(n+1) \times (n+1)}$. The process outputs $ n $ hidden vectors. This is detailed in Section 2 and Algorithm 1.
>
> We have revised Figure 2 and added an additional architecture diagram in Figure 8 of Appendix A for more intuitive illustration.
>
>
> **Weakness2:**
> >If the goal of creating multiple copies is just to make sure multiple depth connections can be modelled parallelly, is creating such copies actually necessary?
>
> We would like to point out that the hidden vector only needs to be duplicated into $n$ copies when it is first input into the network, and the subsequent $n$ hidden vectors are actually different, as shown in Figure 8.
>
> The subsequent response will explain that these n(>1) hidden vectors are the key to making this method work.
>
> >Can't a single copy be used with different residual strengths? The only difference would be in gradient computation , were additional terms for each depth connection would be added to the singular copy.
>
> I guess the approach mentioned by the reviewer refers to our experiment with n=1 in Figure 14. It does not work, and we have included a detailed analysis explaining why this is the case.
>
> We conducted an analysis using the unfolded hyper-connections method, as we did in Section 4.5. We found that when the rate = 1, the unfolded connection graph is fundamentally different from other cases, as shown in Figure 14 in Appendix F.
>
> In Figure 14, compared to HC$\times4$ models, the $\Lambda$-shaped pattern does not appear. Note that HC$\times1$ does not support the $\Lambda$ pattern in its mathematical formulation, in which the connections to previous layers must be either weakened or strengthened simultaneously.
> Thus, the lack of connections from the early layers to the final layers may lead to gradient vanishing, similar to post-norm style transformers, which results in performance degradation. For models with rate ≥ 2, this issue does not arise, resulting in improved performance.
>
>
> >Can an ablation be conducted between DHC (n=4 and n=1) and SHC (n=4 an n=1) to show the importance and need for additional copies.
>
> A corresponding ablation study has been conducted for the DHC method proposed in our paper. Please refer to Figure 5 and Table 1, where the performance of n=1 degrades at 500 tokens but gradually approaches the baseline as training continues. Please also refer to Figures 14 and 15 in Appendix L of the revised version.
>
> Furthermore, we believe this issue is very critical, and we have been thinking about it as well. In fact, in our subsequent research, we have indeed found that it is possible to achieve gains without using n copies, although the gains are smaller. The core idea here is not to copy n times but to split the hidden state into n parts. We will disclose the related results of this research in our future work.
>
> **Weakness3:**
> Thank you for the suggestion; we have carefully revised this figure.

---

> ### Author Response · Authors · 2024-11-21
>
> **Question1:**
> This is a very interesting and profound question. In fact, we conducted a related analysis in Figure 7 and Section 4.5 of the paper. We can study this issue by unfolding the hyper-connections. After unfolding the hyper-connections, we can observe the influence of the hidden state from layer $j$ on layer $i$. In Figure 7, the intensity of the red color represents the magnitude of the influence. In the lower triangular, the uncolored areas represent redundant connections.
> Overall, we observe the following:
> 1. There are fewer redundant connections in the shallow and deep layers, but more in the intermediate layers. Notably, the shallow layers have fewer redundant connections, which **prevents the issue of vanishing gradients**.
> 2. The attention layers exhibit a short-term pattern in their influence on subsequent layers, mainly affecting nearby layers. In contrast, the feedforward network (FFN) has a long-term influence on subsequent layers. It’s important to note that the reduction in **the short-term pattern indicates a lower risk of collapse**.
>
> Additionally, since our initialization is equivalent to Pre-Norm, the connection pattern, as shown in the third diagram of Figure 7, exhibits a fully-connected pattern. This ensures smooth gradient flow in the early stages of training, while dynamically balancing vanishing gradients and representation collapse as training progresses.
>
> **Question2:**
> We consider this proposal very interesting, and we have included this part of the visualization analysis in Appendix E.
> We randomly select three categories from the ImageNet dataset and sample the corresponding examples from the validation set. These samples are fed into the ViT-Base/16-DHC$\times$2 model to compute the dynamic connection weights of the DHC in the final layer. As shown in Fig. 12, we visualize the distribution of these weights. We observe that the intra-class distribution of beta is highly concentrated, indicating that samples within the same category tend to have similar beta values. In contrast, the distribution of alpha is less concentrated, but the differences between the distributions of different categories are more pronounced, as exemplified by $\alpha_{2,0}$.

---

> ### Comment · Reviewer_VJDJ · 2024-12-02
> **Final Comments**
>
> I have edited my review to correct the wrong notion that the algorithm recursively creates n copies. I thank the authors for answering my concerns and questions. I also understood the argument for the need of multiple copies. I don't raise my score to 10 because, the memory footprint is still on the higher end. I also note the public comment and its claims but I think this paper produces a novel contribution and the statement made on DHC only improving because of width to be incorrect (note my strength no 3). Ultimately, I suggest acceptance of the paper and maintain my score of 8.

---

### Official Review · Reviewer_DUBh · 2024-11-03

**Soundness:** 3
**Presentation:** 3
**Contribution:** 3
**Rating:** 6
**Confidence:** 3

**Summary:**

This paper presents hyper-connexions, a new neural network architectural improvement which consists in dynamically adjusting the residual connections between layers, effectively managing the trade-off between vanishing gradients and feature collapse. Many experiments with LLMs and vision models demonstrate the effectiveness of hyper-connections to improve stability of training and downstream performance. Various hyper-connections patterns are also studied in depth, with thorough ablations and visualizations.

**Strengths:**

- There is a clear signal that incorporating hyper-connections in LLMs architectures, without any other modification, improves the training loss for a given number of tokens, and boosts performance on downstream metrics. This result is validated for both dense and MOE architectures.

- Hyper-connexions help reduce training instabilities. This is clear by looking at Figure 5 and 6, the training curves of the models with hyper-connections are smoother and do not have spikes, which is a major advantage for training large models.

- The author did a thorough analysis of the learned connections patterns, with nice visualizations.

- The results generalizes to the vision modalities with experiments on image generation and classification. Hyper-connections seem to be a general improvement for the transformer architecture.

**Weaknesses:**

- The main concern I have with this paper is the computational impact of replicating the activations of the network $n$ times for hyper-connections. There is no study on the computational impact both in terms of running time and memory usage. The authors mention Line 394 that “Both methods expand the hidden size by n times with negligible computational overhead” but it is not shown with a proper experiment on the throughput, overall running time, and peak memory usage. Also, it seems that n=1 performs worse than no hyper-connection, so if n>=2 is necessary, and the memory usage is high, it is necessary to study the trade-off between downstream performance, stability and computational cost.

- Although the signal is promising, a full experiment with scaling the number of tokens beyond 500B will be necessary to fully validate the approach. Not asking for this experiment, but current best LLMs are trained on many more tokens and exhibit much better performance than the number reported. I would be curious to see if hyper-connections are useful for training state-of-the-art LLMs.

- In section 3.2 several non-learnable patterns are presented but are not tried in practice. It is not clear whether learning the hyper-connection patterns is really better that having simple fixed patterns, and an analysis on that would be interesting.

**Questions:**

- Why is expansion rate = 1 worse than no hyper-connection, do you have an intuition ?

- Do these findings generalize to other types of architectures such as ResNets ?

- Line 345 typo: “Static”

---

> ### Author Response · Authors · 2024-11-21
>
> We greatly appreciate the time and effort you have taken to review our manuscript. In response to your insightful comments, we address each point individually.
>
> **Weakness1:**
> We have provided an analysis of the parameter count, computational cost and memory footprint of Hyper-Connections in Appendix B. Since there is still space in the table for the 7B experiments, we have added the parameters and computation (FLOPs) to the 7B experiment table. The detailed data is presented as follows.
>
> ### Table: Comparison of number of parameters
>
> | **Method**          | **HC Params (B)** | **Total Params (B)** | **Total Params Δ rate (%)** |
> |---------------------|-------------------|----------------------|-----------------------------|
> | **OLMo-1B**         | -                 | 1.17676442           | -                           |
> | **OLMo-1B-SHCx2**   | 0.0000026         | 1.17676467           | **+0.00002%**               |
> | **OLMo-1B-SHCx4**   | 0.0000077         | 1.17676518           | **+0.00007%**               |
> | **OLMo-1B-DHCx2**   | 0.0002625         | 1.17702688           | **+0.02230%**               |
> | **OLMo-1B-DHCx4**   | 0.0003940         | 1.17715846           | **+0.03349%**               |
> | **OLMo-7B**         | -                 | 6.88809574           | -                           |
> | **OLMo-7B-DHCx4**   | 0.0013124         | 6.88967027           | **+0.02286%**               |
> | **OLMoE-1B-7B**     | -                 | 6.91909427           | -                           |
> | **OLMoE-1B-7B-DHCx4** | 0.0003940       | 6.91948832           | **+0.00570%**               |
> ### Table: FLOPs per token in forward pass
>
> | **Method**            | **HC FLOPs (G)** | **Total FLOPs (G)** | **Total FLOPs Δ rate (%)** |
> |-----------------------|------------------|---------------------|----------------------------|
> | **OLMo-1B**           | -                | 2.3536              | -                          |
> | **OLMo-1B-SHCx2**     | 0.0010           | 2.3545              | **+0.038%**                 |
> | **OLMo-1B-SHCx4**     | 0.0031           | 2.3566              | **+0.127%**                 |
> | **OLMo-1B-DHCx2**     | 0.0020           | 2.3554              | **+0.076%**                 |
> | **OLMo-1B-DHCx4**     | 0.0049           | 2.3583              | **+0.200%**                 |
> | **OLMo-7B**       | -                | 13.3647             | -                          |
> | **OLMo-7B-DHCx4** | 0.0197           | 13.3844             | **+0.147%**                 |
> | **OLMoE-1B-7B**       | -                | 2.3580              | -                          |
> | **OLMoE-1B-7B-DHCx4** | 0.0049           | 2.3629              | **+0.208%**                 |
>
> The introduction of HC results in a minor increase in activation memory usage during training . This contributes less than **15%** , as we analyzed in Appendx .
> Furthermore , we have developed highly effective engineering optimizations. Since Hyper-Connections introduce very little additional computation, their computational cost is minimal. As a result, during the training phase, memory usage can be reduced through recomputation, while training speed can be maintained by leveraging Triton operators. Based on our current optimizations, we have reached the following conclusions:
> - Training phase: With recomputation and Triton operator optimization, when n=2, peak memory increases by **8%**, and training speed reaches 90% of the baseline.
> - Inference phase: The hidden states generated in each layer can be immediately freed, making the impact on memory usage during inference negligible.
> We will provide the final engineering solutions and detailed numbers in the open-source repository.

---

> ### Author Response · Authors · 2024-11-21
>
> **Weakness2:**
> We appreciate the suggestion to train on more tokens as a means of enhancing the impact of this work. And in fact, we are actively coordinating resources to apply the methodology to the current production-level training of large language models (e.g. models like GPT-4o, Claude, Gemini). However, for academic research endeavors, it is challenging to directly train a model from scratch until convergence, as it would demand an immense amount of computational resources. For instance, Meta's training of the LLaMA-3.2 1B model (https://huggingface.co/meta-llama/Llama-3.2-1B) utilized up to 9T tokens and consumed 370,000 H100-GPU hours, equivalent to 128 H100 GPUs for 4 months or 128 A100 GPUs for 1 year.
>
> And it is crucial to note that for the majority of research related to pre-training improvements, the performance gap between methods stabilizes once a certain threshold of training corpus is reached. To further elucidate this point, we provided the pre-training loss curves up to 500B tokens , as shown in Figure 1 and Figure 6. For some of the 1B model experiments, we extended the training trajectory to even 1T tokens (each experiment requires 64 A100 GPUs for 15 days), where the gains from hyper-connections were consistently maintained. If the reviewers are interested in these additional loss curves, we have included them in Appendix L.
>
> Moreover, based on our past experience in training production-level LLMs, we are confident in the effectiveness of this method, even for model scales with hundreds of billions of parameters.
>
> **Weakness3:**
> In section 3.2 several non-learnable patterns are presented but are not tried in practice. It is not clear whether learning the hyper-connection patterns is really better that having simple fixed patterns, and an analysis on that would be interesting.
> response：
> section 3.2 is  SEQUENTIAL-PARALLEL DUALITY
> We believe this is an excellent suggestion, and we would also like to point out the following:
> 1. The "sequential configuration" is exactly equivalent to the baseline we are comparing against.
>
> 2. As for the "parallel configuration," which is an almost parallel transformer block (PTB), it is commonly used in Google-related models with the primary goal of speeding up inference. This technique is mainly applied to overlap the computation of the FFN (Feed-Forward Network) and memory access in attention, thereby achieving inference acceleration. This is also the reason why the flagship model Gemini 1.5 Pro adopts the "sequential configuration," while only the lightweight model Gemini 1.5 Flash employs this technique. We have tried PTB in LLMs for other projects, and while the training loss can be brought to parity, the reasoning ability significantly deteriorates.
>
> Nevertheless, we believe that the "parallel configuration" is not always inferior to the "sequential configuration" for all problem instances. Therefore, allowing the model to learn and decide which configuration to lean towards based on the input is a reasonable design.
>
> For this work, we have started training the parallel configuration experiments, but the results may not be available until after the rebuttal period. If the paper is accepted, we plan to include them in the camera-ready version.
>
> **Question1:**
> We believe this issue is very critical, and we have been thinking about it as well. We have included our analysis of the rate=1 case in Appendix F.
>
> We conducted an analysis using the unfolded hyper-connections method, as we did in Section 4.5. We found that when the rate = 1, the unfolded connection graph is fundamentally different from other cases, as shown in Figure 14 in Appendix F.
>
> In Figure 14, compared to HC$\times4$ models, the $\Lambda$-shaped pattern does not appear. Note that HC$\times1$ does not support the $\Lambda$ pattern in its mathematical formulation, in which the connections to previous layers must be either weakened or strengthened simultaneously.
> Thus, the lack of connections from the early layers to the final layers may lead to gradient vanishing, similar to post-norm style transformers, which results in performance degradation. For models with rate ≥ 2, this issue does not arise, resulting in improved performance.
>
> Similar experiments and conclusions can be found in Table 1 of https://arxiv.org/pdf/1603.05027: namely, that a shortcut in the residual connection with weights predicted by a network does not perform better than the standard residual connection.
>
> In fact, in our subsequent research, we have found that it is possible to achieve gains without using n copies, although the gains are smaller. The core idea here is not to copy n times but to split the hidden state into n parts. We will disclose the relevant results of this research in our future work.

---

> ### Author Response · Authors · 2024-11-21
>
> **Question2:**
> Theoretically, our method is independent of the model architecture and is compatible with CNNs, as it primarily improves residual connections, which are also present in ResNet.
> Expanding the application of Hyper-Connections to large-scale vision-language models (including CNNs) and text-to-image/video generation tasks is a key direction for our future work. We leave the detailed exploration of these applications to future research.
>
> **Question3:**
> Fixed, thanks.

---

> ### Author Response · Authors · 2024-11-24
>
> We are greatly appreciative of your insightful and constructive feedback, which has been instrumental in elevating the quality of our work. And we remain open to any further questions or concerns you might have regarding our response.
>
> We strongly believe that our approach holds great potential to become increasingly practical and impactful in the era of large language models. We are optimistic that our findings and contributions will provide meaningful value to the community. As the discussion phase progresses, we would greatly appreciate it if you could share any additional thoughts or re-evaluate the paper’s rating at your earliest convenience.

---

> ### Comment · Reviewer_DUBh · 2024-11-25
>
> The rebuttal partially answers my main concern regarding the computational cost of hyper-connexions. Training flops and additional #parameters are not impacted, however memory using is still unclear. Why not showing a Table similar to your two tables for flops and #parameters, but for peak memory usage ? 15% peak memory increase for n=2 is not negligible, what is the increase for n>2 ?

---

> ### Author Response · Authors · 2024-11-25
>
> Thank you for your suggestion. We conducted experiments on 1B models using 8 A100 GPUs to observe their memory usage.
>
> ### Table: Comparison of Memory footprint of 1B models
> | **Method**          | **Peak Memory (G)**    | **Memory Δ rate (%)** |
> |---------------------|-------------------|----------------------|
> | **OLMo-1B**         | 41.11             | -                    |
> | **OLMo-1B-SHCx2**   | 47.55             | **+15.7%**           |
> | **OLMo-1B-SHCx4**   | 51.85             | **+26.0%**           |
> | **OLMo-1B-SHCx8**   | 60.44             | **+47.0%**           |
> | **OLMo-1B-DHCx2**   | 47.56             | **+15.7%**           |
> | **OLMo-1B-DHCx4**   | 51.86             | **+26.1%**           |
> | **OLMo-1B-DHCx8**   | 60.45             | **+47.0%**           |
> It is worth noting that these results are not the optimized version of HC. We plan to release our engineering optimizations in the future.
>
> We hope this addresses your concerns. If you have any further questions, please let us know—we would be more than happy to clarify.

---

> ### Author Response · Authors · 2024-11-26
>
> The complete memory footprint results **without `HC hidden states checkpointing`** are as follows:
>
> ### Table: Comparison of Memory Footprint **Without `HC hidden states checkpointing`** on 8 A100 GPUs
> | **Method**          | **Memory (GB)**   | **Memory Δ Rate (%)** | **Micro Batch Size (tokens per GPU)** |
> |---------------------|-------------------|-----------------------|-------------------------------|
> | **OLMo-1B**         | 41.11             | -                     | 16384                         |
> | **OLMo-1B-SHCx2**   | 47.55             | **+15.7%**            | 16384                         |
> | **OLMo-1B-SHCx4**   | 51.85             | **+26.0%**            | 16384                         |
> | **OLMo-1B-DHCx2**   | 47.56             | **+15.7%**            | 16384                         |
> | **OLMo-1B-DHCx4**   | 51.86             | **+26.1%**            | 16384                         |
> | **OLMo-7B**         | 26.27             | -                     | 2048                          |
> | **OLMo-7B-DHCx4**   | 33.70             | **+28.28%**           | 2048                          |
> | **OLMoE-1B-7B**     | 31.59             | -                     | 4096                          |
> | **OLMoE-1B-7B-DHCx4** | 34.65           | **+9.7%**             | 4096                          |
>
>
> We would like to reiterate that:
> 1. **memory usage is significantly reduced** by leveraging the **`HC hidden states checkpointing technique`**. Specifically, for activations, **only the inputs and outputs of each layer are stored**, while the expanded hidden states are **recomputed during training** through the HC module. This approach not only **minimizes memory consumption** but also **maintains computational efficiency**.
>
> 2. HC is particularly effective for models such as **MoE**, which combine **large parameter counts** with **relatively small hidden sizes**.

---

> ### Comment · Reviewer_DUBh · 2024-11-26
>
> Thank you for the new table on memory usage. I believe the memory usage overhead is reasonable and I understand the argument of activation checkpointing. For these reasons I am increasing my score to 6. Please include the new tables in future versions of the paper.

---

> > ### Author Response · Authors · 2024-11-26
> >
> > Thank you for your valuable feedback, we have added this table to the latest version of the paper.

---

### Official Review · Reviewer_HHYn · 2024-11-10

**Soundness:** 3
**Presentation:** 4
**Contribution:** 3
**Rating:** 6
**Confidence:** 3

**Summary:**

The paper introduces Hyper-Connections, a novel extension to residual connections that dynamically adjusts the strength of connections between layers in deep neural networks. This method addresses the limitations of traditional residual connections, such as gradient vanishing and representation collapse, by introducing Depth-Connections and Width-Connections thus enabling both cross-layer and intra-layer interactions. A dynamic variant of residual connection named Dynamic Hyper-Connections (DHC), further adapts connection strengths based on input. The approach is evaluated extensively across pretraining of large language models (LLMs), Mixture-of-Experts (MoE) models on vision tasks. Experimental results demonstrate significant improvements in training stability, convergence speed, and model performance on various benchmarks, highlighting Hyper-Connections as a general-purpose enhancement for neural architectures.

**Strengths:**

- The paper provides a clear and systematic extension to residual connections named Dynamic Hyper-Connections (DHC), where residual could be consider a static hyperconnection, addressing the trade-off between gradient vanishing and representation collapse.
- Experimenttal results demonstrated effectiveness of DHC across diverse domains, including LLM pretraining and vision tasks.

**Weaknesses:**

- There seem a lack of comparsion to a fullt enabled depth-connections and width-connections (DenseNet style) where all of the connection in Figure 2 are enabled and learnable.
- The main result focus on LLM and downstream peformance on langugage tasks, the result on Vision task in the appendix seem to demonstrate less gain compare to langugage tasks, can the author elebrate more on this?

**Questions:**

- The author mentioned other baselines such as Altup and ResiDual had gains in the early stages of training, Can the author show the full loss curve of OLMo-1B-ResiDual and OLMo-1B-Altup×2 in Table4?

---

> ### Author Response · Authors · 2024-11-21
>
> We greatly appreciate the time and effort you have taken to review our manuscript. In response to your insightful comments, we address each point individually.
>
>
> **Weaknesses1:** It should be noted that our hyper-connections make all connections learnable, as detailed in Section 2. The performance under this setting is compared in Tables 1 and 2 (SHC/DHC).
> The different lines in Figure 2 are for illustrative purposes to correspond to the captions and do not indicate actual connections. We have revised Figure 2 and added an additional architecture diagram in Figure 8 of Appendix A for more intuitive illustration.
>
> **Weaknesses2:** We would like to point out that the visual gains are only relatively smaller compared to LLMs, but they are still significant. Table 7 hyper-connections in DiT performance **comparable to the 1.5 larger model**; Table 8 has **significant improvement from 76.38/77.25 to 77.60/79.94**.
>
> Regarding this phenomenon, we have some intuitive analysis:
> 1. The gain from Hyper-Connections in reasoning ability is hard to manifest in these vision tasks. Hyper-Connections, to some extent, unlock the potential of network depth (alleviating representation collapse), and network depth has a significant impact on reasoning ability (https://arxiv.org/abs/2407.20311). This is why we see a very robust gain on tasks like HellaSwag.
>
> 2. The relatively small scale of the datasets used in these vision tasks may diminish the effect of Hyper-Connections (HC). For example, datasets like ImageNet are smaller compared to those used for large language models, and the training process spans many epochs—for instance, ViT is trained for 300 epochs, and DiT for 1400 epochs. In contrast, large language models typically pass through the data only once, and we observe that the gain from Hyper-Connections does not diminish as the number of training tokens increases. However, in vision tasks, especially in the later stages of training, we notice that the gain from Hyper-Connections tends to decrease as training continues. This may be due to the fact that these vision tasks involve repeated passes over the same dataset across many epochs, which could lead to diminishing returns from the additional capacity provided by Hyper-Connections. Despite these limitations, Hyper-Connections still exhibit notable performance gains in vision tasks. We believe that the full potential of Hyper-Connections may be realized in large-scale vision-language models or in tasks such as text-to-image/video generation, where larger datasets and more complex reasoning may come into play.
>
> Expanding the application of Hyper-Connections to large-scale vision-language models and text-to-image/video generation tasks is a key direction for our future work. We believe that these areas, with their larger datasets and more complex reasoning requirements, provide an ideal environment to further unlock the potential of Hyper-Connections and explore their full capabilities.
>
> We have included the loss curve for training on ViT  in Figure 11, Appendix E. Please refer to it for additional details.
>
>
> **Question1:** Thanks for your suggestion. We have included the loss curves in Figure 13, Appendix L.

---

> ### Author Response · Authors · 2024-11-27
> **A Kind Reminder**
>
> We sincerely appreciate your thoughtful and constructive feedback, which has been invaluable in improving our work. As the deadline for submitting a revised version of the PDF (November 27, AoE) is approaching, we kindly remind you that if there are any remaining questions or suggestions for improvement, we would be more than happy to address them.
>
> Once again, we deeply appreciate your time and effort, and we would be truly grateful if you could re-evaluate the paper’s rating.

---

### Author Response · Authors · 2024-11-21
**General Response**

We appreciate the reviewers for dedicating their time to thoroughly evaluate our manuscript and offering constructive feedback. We are gratified to see the universal acknowledgment of the significance of our work. Incorporating your recommendations, we have meticulously revised the document, with all modifications denoted in red within the updated version. The key changes are as follows:
1. Update Description (in response to reviewers `VJDJ`, `DUBh`) . **We revised the introduction, redrew Figure 2, updated its caption, and added the complete network architecture in Appendix A.** This provides a better introduction to our method.
2. **Efficiency Analysis** (in response to reviewers `fvae`, `VJDJ`, `DUBh`). We added the parameter and computational costs to the 7B experiment table and provided the parameter and computational costs for all experiments in Appendix B. **This clearly demonstrates that the parameter, computational overhead and memory footprint introduced by our method is negligible.**
3. Update Appendix L to encompass the loss curves for altup and ResiDual (in response to reviewer `HHYn`).
4. Update Appendix L to address the concern regarding scalability for results beyond 500B tokens (in response to reviewer `DUBh`). For some 1B models, we extended the training trajectory to even 1T tokens, where the gains from hyper-connections were consistently maintained. Furthermore, based on the magnitude of the reduction in training/validation loss and our prior experience in training production-level LLMs, we are confident in the efficacy of this approach, even for model scales with hundreds of billions of parameters.
5. Update Appendix E to include the training curves of ViT (in response to reviewer `HHYn`),  which explains why the gains in vision tasks are not as significant as those in LLMs.
6. Update Appendix F to explain why the performance is suboptimal when n=1 (in response to reviewers  `VJDJ`, `DUBh`).

We believe that these revisions have significantly strengthened the quality and impact of our work, and we hope that the reviewers will find the manuscript now suitable for publication. We welcome any additional feedback or clarification that the reviewers may have, and we remain committed to addressing their concerns to the best of our ability.

---

### Public Comment · ~Harvie_ZHANG1 · 2024-12-01
**Questions for Hyper-Connections**

Dear Authors,

I have a few questions regarding your proposed Hyper-Connections.

1. **Motivation of the Paper:** The primary goal of residual connections is to minimize information loss at each layer, as referenced in  https://arxiv.org/pdf/2401.17948.pdf. Your proposed framework is similar in that it increases the model's width by copying the hidden features $n$ times.

2. Hyper-connections effectively transform the input using a small number of training parameters. This aligns with the concept of existing Hyper Interaction, as introduced on page 6 of the above paper.

3. The results presented in the paper indicate that only DHC-based models are effective, and their performance is solely enhanced by increasing the model's width. Please refer to my first question.

I look forward to your response if there are any misunderstandings.

Best regards,

Harvie

---

> ### Author Response · Authors · 2024-12-02
>
> Thank you for recognizing our work and engaging in this constructive dialogue with us.
>
> After carefully reviewing the hyperZZW  paper, I would like to clarify the distinctions between our works.
>
> We will now proceed to address each question individually:
>
> > **Motivation of the Paper:** The primary goal of residual connections is to minimize information loss at each layer, as referenced in https://arxiv.org/pdf/2401.17948.pdf. Your proposed framework shares similarities in that it increases the model's width by duplicating the hidden features $n$ times.
>
> We would like to compare **Figure 8** in Hyper-Connections with **Figure 3** in HyperZZW.
> We believe that the differences between this work and ours are significant.
>
> 1. For Hyper-connections, the approach involves duplicating the hidden vector $n$ times **only once** at the beginning of the network. At each layer, a linear combination of the $n$ hidden vectors is used as the layer's input, while maintaining the $n$ hidden vectors with information exchanged through width connections. Then, through deep connections, the layer's output is linearly combined with the n hidden vectors, resulting in $n$ hidden vectors. The coefficients for these linear combinations are scalars.
>
> 2. For HyperZZW, it appears that each SFNE block processes the input $x$ through multi (9) branches (similar to GoogleNet (https://arxiv.org/pdf/1409.4842)), and then concatenates the outputs of these 9 branches.
>
>
> There is a fundamental difference between retaining n hidden vectors throughout the network and repeating the input into n parts for each layer. The former can retain diverse combinations of information from earlier layers across different hidden vectors, while the latter merges the information early on.
>
> ---
> > Hyper-connections effectively transform the input using a small number of training parameters. This aligns with the concept of Hyper Interaction, as introduced on page 6 of the above paper.
>
> We would like to compare **Figure 2 (D)** in Hyper-Connections with **Figure 5** in HyperZZW.
>
> **Hyper-connections** aim to construct both **width connections** (across the $n$ input hidden vectors) and **depth connections** (between the layer output and the processed hidden vectors). The coefficients generated by Hyper-connections involve $n \times (n+1)$ scalars, without any constraints on their value ranges.
>
> In contrast, in the **HyperZZW** paper, the proposed SFNE block processes the input through 9 branches. One of these branches is **Hyper Interaction**, which aims to generate a gating mechanism with the same shape as the input ($B \times C \times H \times W$), where the values are in the range $[0, 1]$. This gate then multiplies with the input to extract useful information. This technique is conceptually similar to https://openaccess.thecvf.com/content_ECCV_2018/papers/Sanghyun_Woo_Convolutional_Block_Attention_ECCV_2018_paper.pdf.
>
>
> Additionally, as proven in Section 3 of our paper, Hyper-connections exhibit the **Sequential-Parallel Duality** property, meaning they can dynamically adjust the sequential or parallel arrangement of layers. Hyper Interaction does not seem to possess this property.

---

> > ### Author Response · Authors · 2024-12-02
> >
> > > The results presented in the paper indicate that only DHC-based models are effective, and their performance is solely enhanced by increasing the model's width. Please refer to my first question.
> >
> >
> > 1. **Static Hyper-Connections** are also effective, as shown in **Table 2** of our latest version of the paper.
> > | **Methods**                                  | **V2 Eval Loss ↓** | **V2 Eval PPL ↓** | **V3 Eval Loss ↓** | **V3 Eval PPL ↓** | **Down Stream Avg. Acc. ↑** |
> > |----------------------------------------------|---------------------|-------------------|---------------------|-------------------|-----------------------------|
> > | OLMo-1B                                      | 2.811               | 18.023            | 2.544               | 14.229            | 62.5                        |
> > | OLMo-1B-SHC×2                                | 2.799               | 17.778            | 2.538               | 14.152            | 63.4                        |
> > | OLMo-1B-DHC×2                                | 2.802               | 17.950            | 2.534               | 14.114            | 63.0                        |
> > | OLMo-1B-SHC×4                                | 2.791               | 17.671            | 2.528               | 14.025            | 63.6                        |
> > | OLMo-1B-DHC×4                                | 2.781               | 17.509            | **2.515**           | **13.826**        | 63.8                        |
> >
> > 2. While our network appears to increase its width, the width of the **FFN** and **Attention** modules remains unchanged. The $n$ duplication at the beginning of the network is primarily for allowing different hidden vectors to retain diverse combinations of earlier-layer information. This helps address the `gradient vanishing` and `representation collapse` trade-off，as outlined in the motivation section of our paper.
> >
> > 3. The $n$ expansion is not strictly necessary, as mentioned in our rebuttal discussion below. We will release a solution that achieves performance gains without this expansion, though the gains are relatively smaller.
> >
> >
> >
> > ---
> > Once again, we sincerely appreciate your interest in our work and the insights you’ve shared about your research. If you have further questions, we would be more than happy to continue the discussion.

---

### Author Response · Authors · 2024-12-04
**Final Author General Response**

We thank all reviewers for their thoughtful feedback and constructive suggestions, which have greatly helped us improve the quality of this work. Below, we summarize the main contributions of our paper, highlight the points of agreement from reviewers, and address key questions and concerns raised during the review process.

---

### **Summary of Contributions**

This paper introduces **Hyper-Connections**, a novel and simple alternative to residual connections that addresses fundamental issues such as **gradient vanishing** and **representation collapse**. The key idea is to dynamically adjust the strength of connections between features at different depths and enable layer rearrangements, improving overall gradient flow and feature representation.

Hyper-Connections achieve notable performance improvements in **LLM pretraining (7B models)**, for both dense and sparse models, as well as in **vision tasks** (including image classification and generation). For the **7B MoE model**, Hyper-Connections achieve a remarkable **1.8x** convergence speedup, and the performance gain remains consistent throughout training. Despite their flexibility, Hyper-Connections introduce negligible computational overhead compared to standard residual connections.

---

### **Points of Agreement from Reviewers**

We are grateful that reviewers acknowledged the following strengths of our work:

1. **Empirical Performance, Generalization, and Stability (`HHYn`, `DUBh`, `VJDJ`, `fvae`)**:
Reviewers widely acknowledged the strong empirical results, particularly the significant improvements in large language model pretraining. Hyper-Connections also demonstrated robust generalization to vision tasks, with experiments on image generation and classification (e.g., ImageNet) showing consistent performance gains. Additionally, Hyper-Connections help reduce training instabilities, as shown in **Figure 5** and **Figure 6**, where the training curves are smoother and free of spikes—an important advantage for training large models.




2. **Visualization and Analysis (`DUBh`, `VJDJ`)**:
Reviewers appreciated the detailed analysis of the learned connection patterns, supported by insightful visualizations. These visualizations reveal the internal logic of the neural network, such as how certain classes follow similar paths compared to others. This provides a deeper understanding of how Hyper-Connections function and their role in addressing issues like gradient vanishing and representation collapse.




---

### **Addressed Reviewer Concerns**

We carefully addressed the primary concerns raised by reviewers, as summarized below:


1. **Parameters, Computational Costs, and Memory Footprint (`DUBh`, `VJDJ`, `fvae`)**:      Concerns were raised regarding whether the flexibility of Hyper-Connections incurs significant **parameters**, **computational costs**, and **memory footprint**. To address this, we supplemented specific numbers for each in our response. The analysis demonstrated that the additional parameters and computational costs are negligible, while the increase in memory footprint is well within a reasonable range. Furthermore, we emphasized that the memory footprint can be further optimized using recomputation techniques. Reviewers appreciated this detailed clarification and acknowledged the efficiency of the proposed method.

2. **Generalization to Vision Tasks (`HHYn`)**:
   One of reviewers questioned the result on Vision task in the appendix seem to demonstrate less gain compare to langugage tasks. We clarified that Hyper-Connections achieve significant performance gains in vision tasks as well. While the gains are relatively smaller compared to LLMs, they remain notable, as shown in **Table 7** and **Table 8**. This highlights the versatility of Hyper-Connections across domains. Furthermore, we offered an intuitive explanation for why the gains in vision tasks are less pronounced, pointing to differences in dataset size and reasoning requirements (e.g., [Physics of Language Models: Part 2.1](https://arxiv.org/abs/2407.20311)).

3. **Conceptual Clarity**:
Reviewers' interpretations of certain concepts in the paper differed slightly from our intended explanation. To clarify these points, we refined our descriptions in the introduction and figure 2 with red highlights to better explain these concepts. These updates provided additional clarity and helped align our presentation more closely with the reviewers' perspectives.

---

> ### Author Response · Authors · 2024-12-04
>
> ### **Conclusion**
>
> We thank the reviewers again for their thorough evaluation and constructive suggestions. We believe that this paper makes important contributions to the design of network architectures by introducing Hyper-Connections, a simple, effective, and broadly applicable alternative to residual connections. Through additional experiments and clarifications, we addressed most key concerns, and reviewers broadly recognized the significance of our theoretical insights, empirical results, and practical utility. We are confident that Hyper-Connections will inspire further research and advancements in deep learning architectures.

---

### Public Comment · ~Harvie_ZHANG1 · 2025-02-09
**Response to authors & Public comments**

Thanks to the authors for their previous responses. Since I didn't receive any reminders, I am now adding more detailed comments.

First, regarding your question about the similarity between my work and GoogleNet (https://arxiv.org/pdf/1409.4842) and CBAM (https://arxiv.org/pdf/1807.06521), I cited and discussed them in my paper. I would also like to clarify that while our approaches are not exactly the same, they share the same underlying principle, although you present it in a different context.

Next, I will outline the similarities between your method and several previous works. **Additionally, you did not discuss the related works mentioned by the Area Chair (FractalNet and DenseNet) in the latest version.**

1. **The use of "Hyper" in neural networks.** As we all know, Hypernetworks (https://arxiv.org/pdf/1609.09106) utilize implicit small networks to generate the weights of the main network, whereas your work employs layer connections as learnable parameters in implicit matrices.

2. **Implicit connections.** DiracNet (https://arxiv.org/pdf/1706.00388) parameterizes network weights as a residual of the Dirac function to eliminate residual connections, and in my Terminator architecture (https://arxiv.org/pdf/2401.17948), this concept is extended to model outputs. Your method can be viewed as a further extension of this idea. Furthermore, DiracNet, as an important variant of residual connections, shares similarities with the initialization of implicit matrices in Hyper-Connections, but you did not discuss this.

2. **Dynamic model weights.** You emphasized the importance of dynamic connections that depend on the input (Section 2.2), which are also referred to as fast weights. Reference links: https://arxiv.org/pdf/2401.17948, https://people.idsia.ch//~juergen/fast-weight-programmer-1991-transformer.html.

3. **Single branch vs. multi-branch.** My work highlights the significance of multi-branch architectures (https://arxiv.org/pdf/2401.17948, visualization results https://github.com/hyperevolnet/Terminator/blob/main/assets/plain_resnet.png) to reduce information loss between model layers. Additionally, multi-branch networks can be traced back to LSTM, and your approach bears similarities to this.

Finally, I have some questions regarding your work.

1. **Motivation:** You use pre-norm and post-norm as motivation, asserting that residual connections cannot effectively address gradient vanishing and representation collapse. For the former, you only cited a paper discussing the existence of gradient vanishing in RNNs (https://ieeexplore.ieee.org/document/279181), while transformers differ in this regard. Moreover, the performance degradation mentioned in the ResNet paper (https://arxiv.org/pdf/1512.03385) is not equivalent to representation collapse (see the third point).

2. **Experiments:** You claim that the proposed Hyper-connections can replace residual connections, but you **only** provide results based on the transformer architecture.

3. **Representation collapse and visualization:** Its definition can be found at https://arxiv.org/pdf/2411.02344, https://arxiv.org/pdf/2406.04267 and https://arxiv.org/pdf/2206.04041, which raises concerns about your visualization result (Fig. 3) and its conclusion.


4. **Ablation study and theoretical analysis:** You did not provide any visualization results or formula derivations demonstrating that Hyper-Connections can alleviate gradient vanishing.

---

### Meta-Review · Area_Chair_t9ou · 2024-12-26

**Metareview:**

The proposed hyper-connections (HCs) are a form of learned architecture that can be optimized to connect different representations across depth and width by summation. In this way they are an extension of residual connections. In addition, HCs can be conditioned on the input to vary these connections during inference for dynamic hyper-connections (DHCs). The main paper concerns the applications of HCs to LLMs, working with open-source OLMoE models, and shows improvements to the loss, data efficiency, and computationally efficiency relative to other recent approaches to expand architectures. In the appendix additional results on vision are shown, where there is still gain, but the effect is more marginal.

Strengths:

- experiments show empirical improvement for pre-training and downstream tasks on text and visual data
- the cost in computation time, memory, and parameters is measured and reasonable (following the rebuttal and revision)
- the idea and the implementation are clear (and have been clarified in the rebuttal phase)

Weaknesses:

- The related work is limited and the scholarship is shallow in its inclusion of only the most recent papers. As a project on connecting across depth and width, there is a well-developed body of work including DenseNets, most practically, and other models such as FractalNets and HighwayNets.
- The work claims similar improvement across text and vision, but more precisely there is a larger improvement for LLMs than current vision models. This is tempered by equally good or still improved results on vision however.
- The organization of the presentation of the paper challenged comprehension, as evidenced by shared questions across reviewers and multiple comments on these points.

Missing: Most critically, the submission is missing discussion (and ideally experiments) for certain related works like DenseNet and it is inadequately organized for ease of comprehension. The issues with exposition however have begun to be addressed in the rebuttal and revision, and could be dealt with in the final revision. Additional discussion is likewise feasible, and would be an acceptable resolution of the related work.

Decision: This work is borderline. The four expert reviewers vote for acceptance (VJDJ: 8, HHYn: 6, DUBh: 6, ) and rejection (fvae: 5). The meta-reviewer sides with acceptance because of the strength of the empirical results for task performance and computation, the satisfaction of the majority of the reviewers and the points addressed in the rebuttal, and the generality of the results across multiple tasks, datasets, and modalities. However, the meta-reviewer cautions that the lack of discussion and experimentation on related work about more sophisticated handling of depth (DenseNets, FractalNets, ...) remains a negative. For broader impact and improved reception of the proposed technique it may be advisable to incorporate more material about these prior works, as has likewise been suggested by the reviewers.

Note: The meta-reviewer acknowledges the confidential comment from the authors. The decision reflects the paper, reviews, rebuttal, and the full discussion.

**Additional Comments On Reviewer Discussion:**

The authors provided point-by-point rebuttals for each review and a general response. Each reviewer engages with the rebuttal and either confirmed or updated their rating. Common points among reviews, for instance questions about the time/memory/parameter cost of the method and criticisms about clarity and organization, were addressed in the rebuttal. More specific points of miscomprehension or requests for additional results were likewise met. Much of the relevant content was included in the appendix—perhaps underlining a need for reorganization or clearer pointers—or provided in the rebuttal. The largely satisfactory outcome of the rebuttal and discussion is shown by the maintained ratings and confirming comments (VJDJ: 8, HHYn: 6) and increased ratings (DUBh: 5 to 6, fvae: 3 to 5).

---

### Decision · Program_Chairs · 2025-01-22

Accept (Poster)